# Semi-Supervised Preference Learning for Multi-modal Large Models via Risk Analysis

## Abstract

Ensuring that multi-modal large models possess reasoning capabilities aligned with human preferences is of paramount importance. Currently, the most effective approach involves fine-tuning these models using reward models optimized towards human-aligned objectives. However, optimization of reward models typically requires large-scale human-annotated datasets, which pose a significant bottleneck for downstream tasks with limited labeled samples. To address this limitation, we propose a **S**emi-supervised **P**reference learning approach based on **R**isk **A**nalysis, denoted by **SPRA**, which can accurately assess the alignment of large model outputs with human preferences using limited labeled data. The proposed SPRA measures preference by a risk model, whose construction consists of three main steps: (1) extract risk features that encode human priors from a limited set of labeled samples; (2) construct a risk model based on risk features; (3) train the risk model. Then, SPRA uses the resulting risk model to rank model responses, with lower-risk ones prioritized as preferred outputs. By explicitly incorporating human priors into its modeling framework, SPRA achieves not only high interpretability but also flexibility to adapt to diverse human preference distributions by adjusting the priors. This contrasts to traditional single-preference predictors, which lack such adaptability. In particular, the SPRA risk model is parameter-efficient, containing only thousands of parameters, which significantly reduces computational overhead and simplifies reward optimization. Our empirical study on real benchmark datasets validates the efficacy of SPRA.

## 1 Introduction

In recent years, the outstanding human-computer interaction capabilities of large language models(LLMs) (Jiang et al., 2023; Guo et al., 2025; Touvron et al., 2023) have attracted significant attention. Consequently, researchers have sought to extend these capabilities to multi-modal large models, as they are better suited to process diverse information found in the real world (Alayrac et al., 2022; Radford et al., 2021; Li et al., 2022).

From this perspective, it is necessary to make the outputs of these large models controllable and usable, i.e., to align their outputs with human preferences. The researchers tackled this issue by incorporating reinforcement learning into both the pre-training and fine-tuning phases of large models. Numerous state-of-the-art methods (Rafailov et al., 2024; Schulman et al., 2017; Wu et al., 2023) have demonstrated the feasibility and efficacy of this approach. They make it easier for large models to perform downstream tasks, such as preference-based learning to transfer models to new tasks (Tang et al., 2025; Lester et al., 2021; Chi et al., 2021) and model self-correction (He et al., 2025). For reinforcement learning, a reward model can be regarded as a preference predictor (Christiano et al., 2017). Typically, a reward model may be either latent or explicit; that is, it can take the form of a constraint equation or a neural network trained on a labeled dataset. In most cases, a high-quality reward model is inseparable from a substantial amount of manually labeled data. However, in practice, labeling a large-scale, task-specific dataset is highly resource intensive.

To reduce the dependence on manually labeled samples, some researchers have resorted to large models to enable automatic annotation of datasets (Bhat & Varma, 2023; Mohta et al., 2024). This

approach generally leverages recent advancements in foundation models to provide AI-generated feedback. For instance, the proposal of ERL-VLM (Luu et al., 2025) employs large VLMs to rate an output into different quality categories, which are then used to generate reward functions for RL agent training. Unfortunately, such approach may not work well in many specific downstream task scenarios, simply because large models that haven't be fine-tuned on particular tasks usually cannot perform high-quality annotation work.

The aforementioned work all underscore the critical importance of reward models. However, AI-generated rewards may still be unreliable for many specific downstream tasks. Therefore, it is very important for preference learning that a reward model can be efficiently trained with a small amount of labeled data from downstream tasks. Towards this aim, this paper proposes a novel risk-based approach for preference ranking. It generally quantifies the relationship between a model's outputs and human preferences by a risk model. The higher the risk, the greater the deviation of an output from human preferences and vice versa. The proposed approach generates multiple outputs on each unlabeled data, which are then ranked by their deviation risk. Lower-risk results are prioritized as preferred outputs (labels). It is noteworthy that in our proposal, the risk model is supposed to be trained on a small set of labeled data. Manual labels can be understood as outputs that conform to human preferences, while the constructed risk model enables autonomous model annotation, effectively saving human labeling cost.

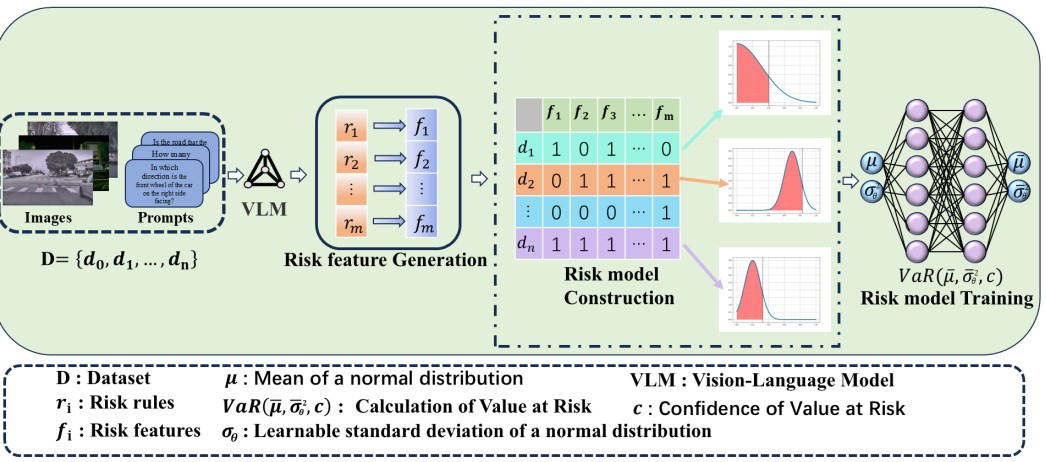

Figure 1: Overview of the SPRA framework: 1) formulating risk risk features by leveraging LLMs and a series of prompts to assess whether the output of the current large model is at risk of errors; 2) The risk features corresponding to each data pair <input, output> are then determined. Subsequently, the means associated with each risk feature are weighted and summed using learnable parameters to yield the final synthesized normal distribution; 3) train the risk model on a small number of labeled data; the risk model prioritizes responses based on risk values.

Specifically, drawing inspiration from existing work on financial risk theory and AI risk analysis, the proposed solution of SPRA consists of three main steps, as sketched in Figure.1: 1) the first step is the extraction of risk features, which are formulated by rules. It first defines a set of general risk rules that mimic human reasoning to constrain the output of the large model. Then, it constructs a binary matrix, referred to as the activation matrix, to evaluate whether the output of a large model satisfies these risk features. In the binary matrix, "1" indicates rule satisfaction while "0" indicates rule violation; 2) in the second step, it uses the generated activation matrix to construct a risk model. In the tasks such as classification or question answering, the answer is unique, and the question and answer form a single data pair. Consequently, the matching probability between the question and the answer can be modeled as a Bernoulli distribution (Chen et al., 2020). Using these distributions, a risk model is constructed by mapping them into a normal distribution based on the activation values; 3) in the final step, a simple neural network is trained on a small set of labeled samples to fit the variance under the normal distribution, allowing us to calculate the risk value for a given confidence.

The efficacy of our proposed approach has solid theoretical grounding, because in the setting of preference learning, higher-quality preference labeled data would generally mean higher performance

for LLM or VLMs. Our proposed approach can automatically construct high-quality preference data with only a limited label information, effectively reducing the cost of manual annotation. In summary, compared with the existing reward models, our proposed SPRA has three key advantages:

- It utilizes only a small number of labeled samples to effectively learn and quantify the relationship between the large model's outputs and human preferences, significantly reducing the burden of manual annotation;

- It measures risk, or preference alignment, by a distribution. Compared with single values predicted by existing reward models, distributions can effectively capture fluctuation risk, whose necessity has been widely accepted by existing risk research;

- It incorporates human-centric priors with a small-sized risk model, making the process of preference finetuning more interpretable and resource-efficient. Compared with existing single reward models, which are usually prone to introducing bias, the proposed risk model can effectively reduce bias by fusing independent outputs from various large models.

## 2 RELATED WORK

**Preference Learning.** RLHF (Christiano et al., 2017) uses human feedback to steer the output of large models, enabling them to better align with human preferences. Subsequent research (Mehta et al., 2023; Ji et al., 2024; Melo et al., 2024) has further built on this approach. Rafailov et al. (2024) argued that the RLHF process is overly complex. Therefore, they propose Direct Preference Optimization(DPO) as a simplified alternative for reinforcement learning. DPO draws inspiration from the Bradley–Terry abstract modeling approach (Bradley & Terry, 1952) for preference optimization. It uses a model trained on a human-labeled dataset as a preference predictor. The large model generates two outputs with different prompts, and these outputs are fed into the preference predictor. The final layer of the predictor produces a probability value, where a higher probability indicates greater alignment with human preferences. A key drawback of this approach is the need for a substantial amount of data to train a robust preference predictor(reward model). Consequently, when the task is changed, a similarly strong predictor must be trained again, which is nearly impossible in tasks with limited labeled data.

Although data-free self-correction and self-rewarding methods are aligned with our goals, they have been shown to be fundamentally limited(Huang et al., 2024; Kumar et al., 2024; Wu et al., 2024b; Kamoi et al., 2024). These works indicate that large-scale fine-tuning is key to achieving stable self-correction or rewarding. Because without extra fine-tuning, a model's abilities stay the same. The simplicity of DPO has spurred further research in preference fine-tuning. Wang et al. (2024b) and He et al. (2025) use DPO to achieve self-correction of the large model. They prompt the model for a second corrective response following its initial reply, then use ground truth as a reference to classify both responses as CHOSEN or REJECTED, thereby constructing a preference dataset for fine-tining the model. It is noteworthy that these proposals did not use an explicit reward model to evaluate the relationship between VLM's outputs and human preferences, but instead used the ground truth as a substitute for the reward model. Unfortunately, given the vast amount of data, obtaining ground truth through manual annotation is prohibitively costly.

Additionally, Park et al. (2022) proposed a semi-supervised reward learning method, approaching preference learning from the perspective of data augmentation to enhance feedback efficiency of preference method. Specifically, they employed a preference predictor to assign pseudo-labels to data for semi-supervised learning. Unfortunately, the quality of these labels remains unknown. Recent advances in foundation model make it possible that a well-trained large model can replace humans in making preference judgments(Li et al., 2023b; Yu et al., 2024; Lee et al., 2024); as a result, a large model can naturally serve as a reward model to provide rewards for a target model(Fan et al., 2022; Rocamonde et al., 2024b). For example, Luu et al. (2025) proposed a framework in which a VLM assigns scores (1, 0) to pseudo-labels generated by the initial policy, and the scoring results are used to construct a dataset for training a reward model in a robot control task, thereby reinforcing the initial policy. This is a standard RLHF(Christiano et al., 2017), where human feedback is replaced with the outputs of the VLM. It can be observed that the efficacy of this approach depends on the capability of VLMs. Similarly, there are numerous methods(Mahmoudieh et al., 2022; Ma et al., 2023; Cui et al., 2022; Sontakke et al., 2023; Rocamonde et al., 2024a) for generating

reward signals through VLMs, including zero-shot learning and contrastive learning. Unfortunately, in many specific downstream tasks such as image classification and VLM Q&A, large models that haven't be fine-tuned on particular tasks usually cannot output high-quality reward signal.

Finally, our work in this paper targets preference learning, we therefore use the mainstream DPO solution for preference fine-tuning(Rafailov et al., 2024). It is noteworthy that our primary contribution is on constructing a preference dataset closely aligned with ground truth using only a limited amount of labeled data, but not on preference fine-tuning methods. Therefore, the existing work on preference fine-tuning methods, e.g., DPO, are orthogonal to ours.

**AI Risk Analysis.** Chen et al. (2020) introduced an interpretable and learnable risk analysis framework for the entity resolution task, and it is called risk analysis. A similar technique, known as confidence ranking, existed before this. The methods (Hendrycks & Gimpel, 2017; Hendrycks et al., 2019; Jiang et al., 2018) based on this technique have garnered attention for their effectiveness in detecting mislabeled data. Due to their lack of interpretability and inability to be task-specific through learning, many subsequent methods (Hou et al., 2018; Sun et al., 2024) have extended risk analysis to various domains, including sentiment recognition and transfer learning. This ranking-based modeling shares certain connections with reinforcement learning. In this paper, we identify the potential of risk analysis as a form of reward modeling, incorporating human awareness as a priori knowledge.

## 3 METHODOLOGY

We define the relationship between the input and output of a vision-language model(VLM) as a matching process. As usual, SPRA treats the matching probability of each data pair as a random variable that follows a Bernoulli distribution with parameter $p$, where $p$ follows a $Beta(\alpha, \beta)$ distribution. The $\alpha$ and $\beta$ are the shape parameters of the $Beta$ distribution. From a statistical perspective, the Beta distribution can be approximated by a normal distribution when $\alpha + \beta \geq 10$. Specifically, in the context of the task scenario, $\alpha + \beta$ represents the total number of samples. Our approach utilizes a set of general rules to capture the risk features of a given dataset. Correspondingly, data with distinct risk features exhibit different matching probabilities. This implies that random variables with varying risk features follow different normal distributions, each with its own mean and variance. This is crucial for accurate risk assessment. In the rest of this section, we will present the technical details of SPRA, including risk model construction & training and risk-based preference finetuning.

### 3.1 RISK MODEL

#### 3.1.1 RISK FEATURE GENERATION

Intuitively speaking, a risk feature serves a predictor judging whether an output matches human preferences. Due to performance unreliablity of large models on specific tasks, we have developed a variety of risk rules that feature each data by means of patterns extracted from human cognition. A risk model would fuse these features to provide accurate and robust risk assessment.

Specifically, we have designed two general types of risk features for the purpose of risk quantification: statistical and generative. The statistical features use the probability distribution of top-n candidates to analyze output risk. In comparison, the generative features analyze risk based on comparing the outputs of large models inferred from different perspectives. Due to space limit, we summarize the designed risk rules(to formulate risk features) in Table.9 in the Appendix. In the rest of this subsection, we illustrate this abstraction process by two examples: image classification and the VLM Q&A.

**Image Classification**. Given an image $i$ and a prompt $x_1$, they are fed into an assistive model $\pi_a$ to get $y_1 \sim \pi_a(y \mid i, x_1)$. The assistive model can be the model used for fine-tuning or another vision-language model. Here, we use the Qwen2VL-2B (Wang et al., 2024a). The $y_1$ is then designed as a new prompt input into a large language model (Lambert et al., 2025) to extract fundamental features $y_1'$ of $y_1$. Finally, a descriptive prompt $x_2$ is constructed to generate $y_2$ through $\pi_a(y \mid i, x_2)$. Determine whether the current data pair conforms to the current rule by following the formula:

$$\text{Result} = \begin{cases} 1, & \text{if BERTScore}(y_1', y_2) \geq \lambda \\ 0, & \text{otherwise} \end{cases} \tag{1}$$

Where 1 indicates that the data pair $(i, x, y_1)$ conforms to the current rule, meaning the data pair is a match. Our implementation uses BERTScore (Zhang* et al., 2020) to calculate the similarity of $y_1'$ and $y_2$. The BERTScore threshold is established to determine the activation state of this rule. To ensure the robustness of risk model, we set the threshold, $\lambda$, to the not overly stringent value of 0.55 in our experiments. .

Figure.2 provides an illustrative visualization of this process. As shown, the human expectation is that the model will output 'dalmatian'. However, the model outputs 'dog', which contradicts human preference. By comparing the features of the object corresponding to the answer with those of the object in the image, it is possible to determine whether the answer aligns with the content of the image.

It is noteworthy that the risk feature illustrated in Figure.2 is only one of generative features we have developed for image classification. In total, we develop 9 risk features for image classification, including both statistical and generative features. For statistical features, we employ the top-k probability distribution of the current token output by the model; for generative features, we compare outputs based on significant prior features among different animal categories. For more technical details on risk features for image classification, pls refer to the Subsection of C in the appendix.

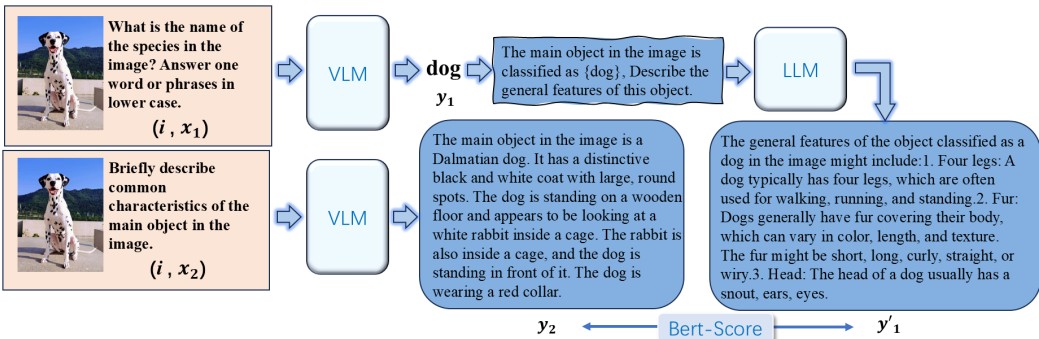

Figure 2: An illustrative example of risk feature for image classification: our preference is "dalmatian"; if the model outputs "dog", the feature descriptions will not match the preference, and the answer "dog" is therefore considered a high-risk response.

**The VLM Q&A.** When humans solve a Visual Reasoning Question, in addition to the information provided in the question and options, they also derive additional useful insights from the image. Therefore, to derive preference information from images, we leverage three models: $M_{des}$ for image description and object detection, $M_{entity}$ for entity extraction, as well as the baseline model $M_{base}$. In the experiment, we select Florence2(VLM) (Xiao et al., 2023) as $M_{des}$, Tulu(LLM) (Lambert et al., 2025) as $M_{entity}$, and Qwen2VL-2B(VLM) (Wang et al., 2024a) as $M_{base}$.

We visualize an example of generattive risk feature for the Q & A in Figure.3. Given an image $i$ along with its corresponding question and options $x$, the baseline model generates an answer, denoted as $a_1$. The image is first described by $M_{des}$. We then used $M_{entity}$ to extract entities from the descriptions of $M_{des}$, questions, and options. These entities are localized in the image through object detection, and the resulting bounding boxes are used to compute the size of each entity, as well as to determine the number of instances of each entity based on the number of bounding boxes. Next, prompts are designed to generate basic information for each entity through $M_{base}$ (e.g., the color of the entity, etc.). We refer to all the information obtained above as "preference information". The preference information is then combined with the question and options, and fed into $M_{entity}$ for answering. Finally, we compare the responses of $M_{entity}$ and $M_{base}$ to determine whether the data pair$(i, x, a_1)$ complies with this rule. A value of 1 is assigned if two responses are the same, indicating the rule is activated; otherwise, a value of 0 is assigned.

Based on the extracted entities, we have also designed questions for each entity to be answered by the baseline model Qwen, such as querying the color or quantity of the entity. Using prior knowledge from the preference information, we evaluated Qwen's responses. Through simple string comparison, we determine whether Qwen's answers are correct. This process is also formulated as a risk feature. Its significance lies in the fact that if a large model can't correctly answer these simple

questions, it indicates that its localization and analysis of entities in the image are essentially wrong. Consequently, this allows us to broadly assess whether its response to the original question poses a risk.

In summary, we have developed 5 risk features for the VLM Q&A task. For statistical features, we similarly perform statistical analysis on the probability distribution of the option tokens generated by the current large model; for generative features, we construct a set of matching problems to evaluate whether the model sufficiently understands the semantics in the image, in order to determine the risk of the model's output. Due to space limit, we have detailed the risk feature prompts and designs for the Q&A task in Appendix Tables.10 and 11.

**Discussion.** A risk feature is supposed to indicate the uncertainty of an output from one perspective. However, using any individual risk feature to describe the risk inherent in a large model's output is usually inadequate. On the other hand, any single risk feature (or reward model) would be prone to introducing bias. Therefore, our proposed approach leverages various risk features, which are supposed to be constructed based on various outputs from different large models. It can be expected that fusing outputs from different large models would effectively reduce preference bias.

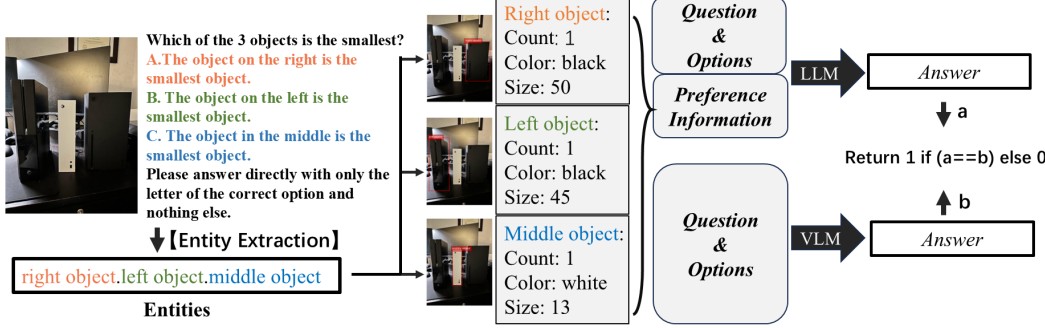

Figure 3: An illustrative example of risk rules for Q&A tasks: first extract the entity information in the image as a known condition of the question for the LLM to answer, and then determine whether the LLM's response aligns with that of the VLM.

### 3.1.2 Risk model construction & Training

**Risk Model Construction:** the construction of the risk model is implemented through the aggregation of distributions that different data follow regarding the probability of matching. Different data exhibit distinct risk features, prompting the generation of an activation matrix based on these variations. The matrix has a shape of $n \times m$, where $n$ represents the number of data and $m$ denotes the number of risk features. Since the matching probability of a data pair follows a Bernoulli distribution with parameter $p$, and $p$ itself follows a $Beta(\alpha, \beta)$ distribution, the mean $\mu_i$ of the matching probability for data pairs sharing the same risk feature $f_i$ can be expressed as:

$$\mu_i = \frac{\alpha_i}{\alpha_i + \beta_i} \tag{2}$$

Where $i$ ranges from 1 to $m$, $\alpha_i$ represents the number of matches in the data with feature $f_i$, and $\beta_i$ denotes the number of non-matches. In practice, the sum of $\alpha_i + \beta_i$ often exceeds 10, so the $Beta$ distribution that the data follows under each feature can be considered a normal distribution. We consider each of the obtained distributions as a risky stock and combine them into one risky stock to predict its value at risk. This aggregation process is achieved through a neural network.

Formally, we define our risk model as follows:

$$\bar{\mu}, \bar{\sigma_\theta}^2 = RiskModel(\mu, \sigma_\theta^2, a) \tag{3}$$

where $\mu$ is the vector comprising the means of the distributions corresponding to each risk feature, $\mu = [\mu_1, \mu_2, ..., \mu_m]^T$; $\sigma_\theta^2$ represents the variance (a learnable parameter), $\sigma_\theta^2 = [\sigma_1^2, \sigma_2^2, ..., \sigma_m^2]^T$; $a$ denotes the the activation matrix, whose corresponding row vector is $a_i = [a_{i1}, a_{i2}, ..., a_{im}]$. The RiskModel in Equation3 is computed as follows:

$$u_\theta = f_\theta(\mu) \tag{4}$$

$$\bar{\mu} = a_i \cdot (w_\theta \circ u_\theta) \tag{5}$$

$$\bar{\sigma_\theta}^2 = a_i \cdot (w_\theta \circ \sigma_\theta^2) \tag{6}$$

where $f_\theta(\cdot)$ is a three-layer perceptron with ReLU as the activation function, $\cdot$ denotes matrix multiplication, $\circ$ denotes element-wise multiplication, and $w_\theta$ denotes the weight corresponding to the risk feature, which is learnable. It can be observed that the aggregation of different distributions is realized by utilizing the cumulativity of normal distributions.

**Risk Model Training:** during the training phase of the risk model, the perceptron $f_\theta$, weights $w_\theta$ and variance $\sigma_\theta^2$ require optimization. According to Equation.6, we obtain the normal distribution corresponding to each labeled data. As in the field of investment theory and existing work on AI risk analysis, we use a popular risk measure, the value-at-risk(VaR), to measure the degree of matching of data pairs, $< input, output >$, i.e. the risk of the model output.

Given a data pair, $d_i$ with a mismatch probability $p$, our objective is to maximize the Var at a confidence level $c$, and conversely, to minimize the VaR when appropriate. This can be expressed as:

$$VaR_c(d_i) = F_i^{-1}(c; \bar{\mu}_i; \bar{\sigma}_i^2) \tag{7}$$

where $d_i$ is a data pair, $F_i^{-1}(\cdot)$ is the quantile function of the normal distribution. $\bar{\mu}_i$ and $\bar{\sigma}_i^2$ denote the mean and variance of the normal distribution that $d_i$ corresponds to. $c$ is set to 90 in our experiments.

Since the risk model cannot be trained directly using VaR, we transform it into posterior probabilities to enable loss calculations with labeled data during the training phase. The loss is defined as follows:

$$-\sigma(VaR_c(d_i))log(y) - (1 - \sigma(VaR_c(d_i)))log(1 - y) \tag{8}$$

where $\sigma(\cdot)$ is Sigmoid function, which is employed to transform VaR into a value within the range [0, 1]. $y$ is risk label, it is derived by comparing the ground truth with the model output. Specifically, the risk label is recorded as 0 when the output of model matches the ground truth and as 1 when it differs.

## 3.2 RISK-BASED PREFERENCE LEARNING

To fine-tune the VLM, we need to construct a preference dataset. During the risk model training phase, we utilized only a subset of the data as the medium for extracting risk features. However, constructing the preference dataset requires the trained risk model to infer on the entire data, which inevitably involves generating risk features for all data. In this section, we discuss how to efficiently generate risk features for all data and the subsequent fine-tuning method. Figure.4 provides an overview of the risk-based preference fine-tuning process. The upper part in the figure illustrates the construction process of preference dataset, where risk model is used to predict preferences, and The lower part depicts the LLM fine-tuning process using the constructed dataset. Note that the process of generating diverse responses with the LLM aligns with the existing SCL construction framework(He et al., 2025).

**Risk Feature Mapping:** in the phase of risk model training, we only need to generate risk features for a small amount of labeled data. However, to apply the trained risk model on unlabeled data, it becomes necessary to input risk features for the unlabeled portion of the data. The risk feature generation process may be very time-consuming, especially when dealing with large datasets. Hence, we present an approximation alternative to use the obtained activation matrix mapped to the unlabeled data.

According to Equation.6, the optimization of the risk model depends on the activation matrix, which captures the relationships between data pairs and risk features. This implies that the risk features can be indirectly described by leveraging the known activation matrix and mapping it to the unlabeled data. By doing so, we save the need to re-generate risk features for the unlabeled data. Specifically, we denote the unlabeled data as $t$ and the activation matrix has been generated as $A$. Each row of $A$ represents the activation vector corresponding to the labeled data, $d$. Subsequently, we employ

Figure 4: Overview of risk based preference finetuning: 1) after the VLM generates the initial response $R_i$, a correction prompt is used to produce a refined response $R_r$; 2) the risk model evaluates both, calculating their value-at-risks, $R_i$ and $R_r$; 3) based on $p_i$ and $p_r$, the preferred response is determined, and fine-tuning is performed using DPO.

the vision encoder in CLIP (Radford et al., 2021) to conduct feature extraction for $t$ and $d$, denoted as $F_t$ and $F_d$. Where the shape of $F_t$ is $B \times 768$ and the shape of $F_d$ is $n \times 768$. The correlation between the two is computed using the following equation:

$$C = F_t \cdot F_d^T \tag{9}$$

where correlation matrix $C$ has a shape of $B \times n$, $B$ is the batch size of unlabeled data, and $n$ is the number of activate vectors, it is also the number of labeled data. Finally, the activation matrix mapping for the unlabeled data is achieved through the following equation:

$$\bar{A} = Softmax(C) \cdot A \tag{10}$$

where A has a shape of $n \times m$.

The reason this mapping method works is that we only need to get the normal distributions corresponding to the data pairs, without having to consider the details in the images.

**Preference Finetuning:** referring to the approach of (He et al., 2025), the data structure of the preference dataset we need to construct is $< r_1, r_2 >$, which represent the first response of the model and the second response given the correction prompt, respectively. To quantify the risk model's capability in preference prediction, we devised a formula for preference prediction accuracy:

$$\frac{\sum_{i=1}^{N} f(r_{i1}, r_{i2})}{N} \tag{11}$$

$$f(r_{i1}, r_{i2}) = \mathbb{I}\left[(p(r_{i1}) > p(r_{i2})) \wedge (r_{i1} = \text{rejected} \wedge r_{i2} = \text{chosen})\right] \\ + \mathbb{I}\left[(p(r_{i1}) < p(r_{i2})) \wedge (r_{i1} = \text{chosen} \wedge r_{i2} = \text{rejected})\right] \tag{12}$$

where $\mathbb{I}[\cdot]$ is the indicator function, which returns 1 if the condition is satisfied, and 0 otherwise. $p(\cdot)$ denotes the posterior probability of the value at risk. As usual, we fine-tune the benchmark model through DPO (Rafailov et al., 2024).

## 4 EMPIRICAL EVALUATION

This section evaluates the efficacy of the proposed SPRA by an empirical study on real benchmark datasets. Since the major contribution of SPRA is using a small amount of labeled data to automatically annotate a large volume of unlabeled data, which can then be leveraged to fine-tune preferences of VLMs. Therefore, our experiments have two primary objectives: 1) to demonstrate that the preference labels generated by SPRA are considerably more accurate than the existing alternatives of large reward models; 2) to demonstrate that using only a limited amount of ground-truth labels, SPRA can achieve highly competitive preference learning compared to the classical preference fine-tuning method of SCL using all the ground-truths.

In the empirical study, we simulate real-world task settings by leveraging existing labeled datasets, where a small portion of the data is designated as labeled and the rest is treated as unlabeled. We primarily validate our approach on the classification and question answering tasks. Specifically,

the classification task involves the Animals dataset (Xian et al., 2018), while the question answering tasks include RealWorldQA (xAI, 2024), SeedBench (Li et al., 2023a), MMBench (Liu et al., 2024), MMStar (Chen et al., 2024), and ScienceQA (Lu et al., 2022). The proportion of labeled versus unlabeled data is adjusted based on the total number of samples in each dataset. Specifically, we use 2% of the data in the Animals dataset as labeled examples for risk model training. In contrast, due to the much smaller size of the question answering datasets, we uniformly allocate 40% of the labeled data for risk model training. In our experiments, to mitigate the randomness inherent in large model generation and risk model training, we report the performance results averaged over five independent tests on each dataset, using identical generation configurations. The detailed risk model parameters, training configurations, and large model generation configurations in the experiments are provided in the Appendix.A.

## 4.1 EVALUATION ON PREFERENCE PREDICTION ACCURACY

Since the existing semi-supervised preference learning approaches mainly leverage multi-modal large models to predict pseudo-labels on unlabeled preference data, we compare the preference prediction capability of our risk model with several SOTA vision-language models. The comparative results are presented in Table.1, in which we report the mean and standard deviation of the outcomes from five runs. It can be observed that risk models perform considerably better than the existing alternatives of large models. On the task of image classification, risk model is particularly well-suited for preference prediction, as the feature representations across different classes are more distinct and the resulting risk features exhibit strong discriminative power. On the Q & A tasks, the advantages of SPRA are also considerable, even though the margins are smaller. The performance of SPRA is impacted by limitations in the tool models used within the risk rules and the inherent complexity of the data.

Table 1: Comparative evaluation of preference prediction accuracy (averages±standard variations).

| | Animals(%) | RealWorldQA(%) | MMBench(%) | MMStar(%) | SeedBench(%) | ScienceQA(%) |
|---|---|---|---|---|---|---|
| Idefics2-8B (Laurençon et al., 2024b) | 45.51(±0.90) | 56.29(±2.75) | 59.29(±1.24) | 51.27(±1.86) | 62.03(±0.38) | 62.46(±1.79) |
| Llava1.5-7B (Liu et al., 2023) | 51.53(±0.06) | 35.33(±3.15) | 50.29(±1.55) | 45.01(±0.98) | 44.68(±0.38) | 41.31(±2.16) |
| MiniCPM-V 2.6 (Yao et al., 2024) | 55.27(±0.26) | 62.63(±2.15) | 66.82(±1.50) | 54.15(±2.04) | 56.59(±0.11) | 68.76(±1.16) |
| **SPRA(Ours)** | **78.73(±1.01)** | **70.68(±3.00)** | **77.43(±1.74)** | **66.79(±1.66)** | **69.89(±1.18)** | **72.17(±1.85)** |

## 4.2 EVALUATION ON PREFERENCE FINE-TUNING

Table 2: Performance evaluation of SPRA preference fine-tuning (averages±standard variations).

| Models | Animals(%) | RealWorldQA(%) | MMBench(%) | MMStar(%) | SeedBench(%) | ScienceQA(%) |
|---|---|---|---|---|---|---|
| Qwen2-VL-2B | 68.24(±0.08) | 48.31(±1.21) | 72.10(±0.35) | 37.84(±0.66) | 60.74(±0.18) | 62.63(±0.65) |
| **Qwen2-VL-2B+ft** | **69.92(±1.61)** | **49.67(±1.53)** | **74.10(±0.49)** | **38.27(±0.61)** | **62.41(±0.90)** | **63.84(±0.21)** |
| Idefics3-8B | 73.17(±1.20) | 60.39(±1.08) | 79.87(±2.15) | 43.67(±1.17) | 66.81(±2.30) | 85.47(±1.21) |
| **Idefics3-8B+ft** | **74.27(±1.11)** | **60.91(±1.15)** | **79.96(±1.16)** | **43.73(±1.04)** | **66.90(±1.11)** | **85.91(±0.98)** |

In this section, following the approach of SCL proposed by (He et al., 2025), we construct a preference dataset and then fine-tune the model using Direct Preference Optimization (DPO). However, unlike the original SCL, which relies on ground-truth labels to determine whether a model output aligns with human preferences, our SPRA leverages the risk assessment provided by the risk model. Furthermore, to demonstrate the efficacy of SPRA, we have also implemented a version of SCL using the same amount of labeled data as SPRA for comparative purpose. It is noteworthy that while comparing SPRA with SCL, they use the same fine-tuning method of DPO.

To evaluate the efficacy of SPRA for preference fine-tuning, We fine-tune both small and large benchmark models, the small Qwen2-VL-2B (Wang et al., 2024a) and the relatively larger Idefics3-8B (Laurençon et al., 2024a) with DPO on the preference dataset constructed by SPRA. The detailed evaluation results are presented in Table.2. All the reported results are the averages and standard variances over five runs. It can be observed that SPRA effectively improves performance, with various margins on different datasets. With a higher mean accuracy and a standard deviation close to that of the baseline, the improvements are not artifacts of random variation but reflect real gains.

Table 3: Comparison of relative performance improvement of fine-tuned models (SPRA vs SCL): 1) SCL-100% leverages all the ground-truth labels, SPRA uses only a small amount of ground-truths and SCL-40% uses the same amount of ground-truths as SPRA; 2) SPRA achieves highly competitive performance compared with SCL-100%, and performs better than SCL-40%.

| Methods | RealWorldQA(%) | MMBench(%) | MMStar(%) | SeedBench(%) | ScienceQA(%) |
|---|---|---|---|---|---|
| Qwen2-VL-2B+SCL-100% | **4.31** | 0.76 | **1.27** | 1.22 | **2.51** |
| Qwen2-VL-2B+SCL-40% | 0.12 | $-$**2.41** | 0.03 | 0.33 | 0.51 |
| **Qwen2-VL-2B+SPRA** | 2.81 | **2.77** | 1.14 | **2.75** | 1.93 |

It is worth noting that our experiments only applied DPO to the model without performing supervised fine-tuning(SFT), in order to align with the experimental results in SCL. With only preference learning, pretrained large models are inherently difficult to improve significantly. For instance, in verifiable tasks (e.g., Q&A), leading optimization methods like SimPO(Meng et al., 2024) and $\beta$-DPO(Wu et al., 2024a) have demonstrated that an improvement of nearly 3% is typically considered effective. Furthermore, preference learning offers more than just advantages in verification accuracy; its primary goal is to align LLM outputs with human preferences. Because large models may make mistakes both before and after preference fine-tuning—yet the nature of these errors differs fundamentally. For instance, in the Animals classification task, one of the labels is "dalmatian." Given the prompt "What is the main object in the image?", the model without preference fine-tuning simply outputs: 'dog'. In contrast, the fine-tuned model responds: "dog, specifically a dalmatian, because it has black and white spots on its body." Although the fine-tuned model's prediction still does not exactly match the label, its output is more consistent with human reasoning and demonstrates better readability.

While these results shown in Table.2 confirm the validity of the preference dataset constructed by SPRA, they do not validate its advantage over the straightforward implementation using the available labeled data, nor do they quantify the quality gap between it and the preference dataset based on ground-truth labels. To provide a more direct comparison, we adopt the method proposed by (He et al., 2025), which leverages ground-truths to compare two responses of VLM and construct a preference dataset, referred to as SCL in their work. As mentioned above, we have implemented two versions of SCL, SCL-100%, which uses all the ground-truths in the training data of benchmarks, and SCL-40%, which uses the same amount of labeled data as SPRA.

The detailed evaluation results on Qwen2-VL-2B are presented in Table.3. Firstly, it can be observed that the performance of SPRA is highly competitive compared with SCL-100%. It can even outperform SCL-100% on some datasets, e.g., MMbench, SeedBench. This is because when constructing the preference dataset, the ground truth filters out all noise, which may cause the optimized model's generalization ability to decline. For the risk model, the diversity of risk rules makes its judgment of preferences less absolute than the ground truth, thereby enriching the content of the constructed preference data. Secondly, it can be observed that if using only 40% of the ground-truth in the training dataset, the performance of SCL is consistently worse than SPRA. This suggests that leveraging a small set of labeled examples to annotate a large amount of unlabeled data is indeed meaningful, as it leads to more substantial performance gains.

## 5 CONCLUSION AND LIMITATION DISCUSSION

This paper introduces a novel semi-supervised preference learning approach of SPRA based on risk analysis for multi-modal large models. Our empirical study has demonstrated that the preference labels generated by SPRA are considerably more accurate than the existing alternatives of reward models. Using only a limited amount of ground-truth labels, it can achieve highly competitive preference learning compared to the alternative of using all the ground-truths.

As a general preference learning approach, the proposed solution can be potentially applied to other tasks and LLM models. However, the proposed solution has the following limitation worthy of future investigation. We have provided a general guideline for risk feature generation; however, the process of risk feature design still involves some manual work. Due to the limited number of risk features, the required manual work is negligible compared with that of manual preference labeling. In future, it is interesting to investigate how to automate and optimize the process of risk feature generation given a downstream task.

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

## A    IMPLEMENTATION DETAILS

Our experiments are implemented using PyTorch (Paszke et al., 2019) and conducted on two RTX A6000 GPUs. For fine-tuning Qwen2-VL-2B, we employ full-parameter fine-tuning, whereas Idefics3-8B utilizes LoRA (Hu et al., 2022). The detailed configurations are provided below.

### A.1    RISK MODEL

We divide the task data into a small portion of labeled data and a larger portion of unlabeled data. The small portion of labeled data is used for generating risk features. The specific division ratio is as follows: Animals(2%), RealWorldQA(40%), MMBench(40%), SeedBench(40%), ScienceQA(40%), MMStar(40%). The division ratio depends on the total number of samples. If the total number is large, the ratio can be appropriately reduced; conversely, if the total number is small, the ratio can be increased.

During the risk model training phase, the appropriate number of epochs can be selected by observing the rate of change in the loss values. In our experiments, the specific training parameters are as follows: batch size is set to 32, learning rate is 0.001. The training epochs: Animals(250 epochs), RealWorldQA(250 epochs), MMBench(250 epochs), SeedBench(250 epochs), ScienceQA(250 epochs), MMStar(250 epochs).

### A.2    FINETUNE

Our DPO code is based on Huggingface (Huggingface). When fine-tuning Qwen or Idefics3, several common parameter settings are typically employed. The learning rate is set to $5e^{-6}$, batch size is set to 1.

**Qwen2-VL-2B.** When fine-tuning Qwen, we observe that training the model on a combined dataset (Realworld + Seedbench + MMStar + MMBench + ScienceQA) leads to a bias in inference capabilities toward tasks with larger sample sizes. To address this, we performed fine-tuning separately for each task to evaluate inference performance. The number of epochs is uniformly set to 3. In the DPO configuration, the $\beta$ is set to 0.4.

**Idefics3-8B.** When fine-tuning Idefics3, we utilize a combined preference dataset, as its trainable parameters are substantial and do not exhibit bias. To optimize memory usage, we employe LoRA (Hu et al., 2022) during this process. The LoRA implementation is based on Huggingface, with default LoRA settings applied. We fine-tune Idefics3 for 10 epochs. In the DPO configuration, the $\beta$ parameter is set to 0.1.

## B    PREFERENCE GENERATION

We evaluated two prompts designed for generating correction responses. The first prompt yielded the results presented in Table.1, while the results of the second prompt are displayed in Table.4.

Table 4: Comparison of preference prediction accuracy under alternative prompt.

| Methods | RealWorld | MMBench | MMStar | SeedBench | ScienceQA |
|---------|-----------|---------|--------|-----------|-----------|
| Llava1.5-7B | 0.40 | 0.61 | **0.55** | 0.54 | 0.53 |
| Qwen2-VL-2B | **0.52** | **0.65** | 0.54 | **0.57** | **0.58** |

The results in Table.4 are notably lower than those in Table.1. However, the risk model's preference prediction capability continues to outperform other methods. We posit that variations in prompts lead to differences in model outputs, thereby affecting the proportions of chosen and rejected responses. Table 6 presents these findings. When using Prompt 1 to generate responses with Qwen2-VL, followed by preference prediction using reward models such as the risk model or LLaVA, the table reports the number of instances corresponding to two scenarios: (1) the first response is chosen and the second is rejected, and (2) the first response is rejected and the second is chosen. Our analysis indicates that the reward model achieves higher prediction accuracy with Prompt 1. This is

attributed to the prevalence of (chosen, rejected) pairs in the data, suggesting that the baseline model frequently produces errors when generating corrective responses.

Table 5: Comparison of the reward model's choice or rejection of results generated by different prompts.

| | Prompt 1 | | | | |
| --- | --- | --- | --- | --- | --- |
| Methods | RealWorld | MMBench | MMStar | SeedBench | ScienceQA |
| Llava1.5-7B | (102, 208) | (862, 1209) | (279, 418) | (3531, 5908) | (441, 600) |
| Ours | (135, 121) | (1098, 540) | (333, 237) | (5632, 2734) | (502, 266) |
| | Prompt 2 | | | | |
| Methods | RealWorld | MMBench | MMStar | SeedBench | ScienceQA |
| Llava1.5-7B | (84, 209) | (527, 864) | (212, 382) | (2284, 4230) | (284, 537) |
| Ours | (132, 169) | (471, 465) | (261, 274) | (2756, 2104) | (349, 293) |

Table.6 provides the prompt for the Qwen2-VL-2B in the Q&A task. Using this prompt, Qwen generates the initial response. Subsequently, we employ the prompt in Table.8 to instruct Qwen to revise its response. Although the risk model demonstrates high accuracy in predicting preferences for responses generated by Prompt 1, as shown in Table.8, we opted to use responses generated by Prompt 2 to construct the preference dataset. We believe that the results produced by Prompt 2 are more suitable for fine-tuning and yield more consistent and well-structured responses.

For the classification task, we designed a relatively straightforward prompt, which explicitly emphasizes our desired preference: the output must be in lowercase and consist of a single word or short phrase. Details are presented in Table.13.

## C  RISK RULES AND PROMPTS

The risk rules are categorized into statistical and generative types. The statistical type derives results through calculations, whereas the generative type relies on prompting a large language model to produce results. We summarize the risk rules used in this paper in Table.9.

We designed multiple rules to evaluate the risks associated with large-model outputs. In the risk rules we define, if f the input is abnormal or the image is ambiguity, most risk rules become ineffective. For example, object detection within the rules and question-answering based on detection results. Most of our rules are based on the primary objects within an image. A ambiguous image indicates that the primary object recognition has failed, resulting in most risk features being set to 1. This consequently increases the likelihood that the risk model will output a high-risk result.

### C.1  CONFIDENCE LEVEL

We treat the probability of model outputs under n-shot as a risk rule. After converting the model's logits into probabilities via the softmax function, if the highest probability value significantly exceeds the others, the large language model is considered highly confident in its response. Conversely, if the highest probability value is only marginally greater than the others, the output is deemed a high-risk response. The criteria for quantifying the magnitude of the probability gap are as follows:

- **Score ratio.** We select the top n probability values corresponding to the logits, with n set to 5 in our experiments. Starting from the smallest of these n values, we divide it by each of the other values, filter out results greater than or equal to 1, and compute the mean of the remaining values to obtain the final result. If the final result is greater than 0 but less than 0.4, it indicates that the model is confident in its current output, as the highest probability significantly exceeds the remaining probabilities. In fact, the standard threshold for the final result is greater than 0 and less than 0.5. However, to ensure a degree of generalization, we adopted a stricter range of greater than 0 and less than 0.4.

- **Entropy.** We use entropy to indicate that the top n probabilities corresponding to the logits of a large LLM's output are closely similar. Here, n is also set to 5. First, the top n largest

Table 6: Prompt used for generating initial response of QA task.

**Responses generation**

**Initial prompt:** Given the question and the corresponding options below, first provide the correct answer. After that, explain step by step why this answer is the best choice, considering the information available. Finally, wrap the correct answer in ## ##.

Example:
Question:
Which of the following is an example of a mammal?
A. Shark.
B. Whale.
C. Lizard.
D. Frog.

Output:
B
Reason:
1.Mammals are vertebrates that typically have hair or fur and give live birth (except for monotremes).
2.Whales, although aquatic, are mammals because they give live birth, nurse their young with milk, and have a warm-blooded metabolism.
## B ##

Question:
What is the chemical symbol for gold? A. Au. B. Ag. C. Fe. D. Hg

Output:
A
Reason:
1.The chemical symbol for gold is derived from its Latin name "Aurum."
2.Au is the symbol that corresponds to gold, making it the correct answer.
## A ##

Question:
{question}

Output:

Table 7: Prompt for matching

**Matching**
Here we have the key information from the image:
{information}
The information is a dictionary whose keys are the entities in the image and whose values represent the current number of entities and their colours. Please answer the following question based on the image information:
{question}

probabilities are normalized, and the entropy is calculated using the natural logarithm (ln) to obtain the final result. The result $x$ should range between 0 and 1.6094. We set a threshold of 1.554. If $x > 1.554$, it indicates that the top n probabilities are relatively close, with no significantly dominant value.

- **Uniformly.** This rule was established to more rigorously assess whether the top n largest probabilities of the model output are uniformly distributed, indicating a lack of confidence in the current output. It can be formulated by combining the two preceding rules. For instance, the Score Ratio evaluates whether there is a dominant probability value, while Entropy assesses whether the output probabilities are uniformly distributed. By taking the inverse of the Score Ratio and combining it with Entropy, we can jointly evaluate the uniformity of the top n probabilities of the model output. This approach enhances the diversity of the generated risk features.

Table 8: Tow correction prompts uesed for refining the initial response.

**Responses generation**

**Prompt 1:** Review your previous answer and ensure that all relevant aspects of the image have been considered. Are there any elements or details that you missed? Based on your review, improve your answer. Your final answer should be put between two ##, like ## A ## (if your final answer is A), at the end of your response.

**Prompt 2:** Review your previous answer and ensure that all relevant aspects of the image have been considered. Are there any elements or details that you missed? Based on your review, improve your answer. First you should output the corrected option. After that, explain step by step why this answer is better than the previous one. Finally, wrap the correct answer in ## ##.
Example:
A
Reason:
1. The reason1.
2. The reason2.
## A ##

Output:

Table 9: Risk rule list. The risk rules proposed in this paper are categorized into two types: statistical and generative. Statistical rules are used to assess the probability distribution of large model outputs, while generative rules are employed for conditional matching.

| Task | Risk Rule | |
| --- | --- | --- |
| | Statistical | Generative |
| Classification | 1.Score ratio 2.Entropy 3.Uniformly | 1.Is animal? 2.Where live? 3.Image feature 4.Descriptive similarity 5.Judging the species 6.Judging the species 2 |
| Q&A | 1.Score ratio 2.Entropy 3.Uniformly | 1.Object count and color 2.Object matching |

We obtained a more diverse set of probability distribution features using few-shot prompted outputs. Subsequently, we evaluated the model's confidence in the current response using the aforementioned method. In our experiments, we utilized 4-shot prompting. Although these three rules are all designed to assess the uniformity of probability distributions, the differences in quantification criteria bring greater diversity. Using different standards provides more tolerance for errors, which in turn enhances generalization.

## C.2 RISK RULES FOR CLASSIFICATION TASKS

Given the classification result $C$ obtained from a vision-language model, we now evaluate it using multiple risk rules, ultimately combining them into a risk feature. We summarize these in Table.10.

Table 10: Risk rules for Classification Tasks.

**Risk rule: Is animal?**
**LLM prompt:** Is {C} an animal? 1(yes) 0(no):
**VLM prompt:** Is the object in this image an animal? If yes print 1, if no print 0:

**Risk rule: Where live?**
**LLM prompt:** Where does {c} live, you can answer by choosing one of (land, water, sky).
**VLM prompt:** Do the objects in this picture live in water, on land, or in the sky? Your final
answer should be put between two ##, like ## sky ## (if your final answer is sky), at the end
of your response.

**Risk rule: Image feature**
**VLM prompt:**
Answer the following questions:

1. Is this image from an anime/cartoon style or a realistic setting?
2. What is shown in this image, including the main objects and more details?

Please ensure each answer is comprehensive and concise, while addressing the specific question being asked.

**LLM prompt:**
{VLM output}
It's part of a description of a particular picture which states whether the picture is from a comic
or reality. By the source of the picture you, then do you think his description makes common
sense? Output 1 (yes) 0 (no)

**Risk rule: Descriptive similarity**
**VLM prompt:** Briefly describe common characteristics of the main object in the image.
**LLM prompt:** The main object in the image is classified as {C}, Describe the general features of this object.

**Risk rule: Judging the species**
**VLM prompt:** Describe the physical appearance of the main object in this image in detail.
Provide the features only, without extra explanations.
**LLM prompt:** Based on the given general features of the object: {C} Can you determine its
exact species or breed? Answer only yes or no.

**Risk rule: Judging the species 2**
**VLM prompt:** Look at the object in the image. Describe its distinctive physical characteristics,
including size, color, shape, and any unique features that distinguish it from similar species.
Avoid general terms, be as specific as possible.
**LLM prompt:** Based on the features of the object: {VLM output} Do all {C} have these
characteristics? Anwser only yes or no.

## C.3 RISK RULES FOR Q&A TASKS

First, describe the input image, and then extract entities from the description. We employ an integrated large-scale model (Xiao et al., 2023) as a tool to describe the image. This step does not involve a specific prompt but is implemented using the identifier '< $MORE\_DETAILED\_CAPTION$ >'. Using the prompt in Table 11, the LLM extracts entities from the description, with the output entities concatenated by periods.

Table 11: The prompt used for extracting entities from image descriptions.

**Entity Extraction from Image Descriptions**

**Prompt:** Given a sentence, extract the entities within the sentence for me.
Extract the common objects and summarize them as general categories without repetition, merge essentially similar objects.
Avoid extracting abstract or non-specific entities.
Extract entity in the singular form. Output all the extracted types of items in one line and separate each object type with a period. If there is nothing to output, then output a single "None".

Examples:
Sentence:
The image depicts a man laying on the ground next to a motorcycle, which appears to have been involved in a crash.

Output:
man.motorcycle

Sentence:
There are a few people around, including one person standing close to the motorcyclist and another person further away.

Output:
person.motorcyclist

Sentence:
No, there is no car in the image.

Output:
car

Sentence:
The image depicts a group of animals, with a black dog, a white kitten, and a gray cat, sitting on a bed.

Output:
dog.cat.bed

Sentence:
The image shows that polar bears and german shepherds.

Output:
polar bear.german shepherd

Sentence:
There are some Smart TVs and electric cars in the image.

Output:
smart tv.electric car

Sentence:
{sentence}

Output:

After first step, entities from both the question and options are also extracted to prevent omitting critical entity information. The results of this step are similarly concatenated with periods. The details are as follows:

**Entity Extraction from Q&A**
Prompt: Given Question and its correspond answer options, extract the entities that can solve the problem based on the combination of the question and options.
Extract entities in singular form. Output all the extracted item types in one line, separated by a period. If there is nothing to output, return "None".

Examples:
Question:
Which direction are the knobs currently in?
A. The knobs are currently in the downward facing position.
B. The knobs are currently in the right facing position.
C. The knobs are currently in the upward facing position.
Please answer directly with only the letter of the correct option and nothing else.

Output:
downward facing knob.right facing knob.upward facing knob

Question:
What color is the closest cup to the camera?
A. Green.
B. Blue.
Please answer directly with only the letter of the correct option and nothing else.

Output:
green cup.blue cup.camera

Question:
Is the large tree to the left or right of the house?
A. Left.
B. Right.
Please answer directly with only the letter of the correct option and nothing else.

Output:
tree.house

Question:
What is the shape of the object on the table?
A. Square.
B. Circular.
C. Triangular.
Please answer directly with only the letter of the correct option and nothing else.

Output:
square object.circular object.triangular object

Question:
What is the object next to the door?
A. Book.
B. Shoe.
C. Bag.
Please answer directly with only the letter of the correct option and nothing else.

Output:
book.shoe.bag

Question:
{question}

Output:

Using the extracted entities as objects to query the VLM. Determine whether large models can identify the content of entities within images through the prompt in Table.12.

Table 12: Prompts for asking count and color.

| | |
|---|---|
| **Entity count** | |
| How many {entity} in this image? Answer only the number. Do not include any additional details or explanations | |
| **Entity color** | |
| What is the color of {entity} in this image? Answer only color, nothing else. If more colors are present, join the results by a period. Like: red.blue.black | |

Use the extracted entity count, colors, and other information as prompts to let the LLM solve the problem, and verify whether the answers given by the vision-language model (VLM) and the large language model (LLM) are consistent. The defined prompts are listed in Table.7.

Table 13: Prompts uesd for Classification.

| |
|---|
| **Initial prompt** |
| What is the name of the species in the image? Answer one word or phrases in lower case. |

| |
|---|
| **Correction prompt** |
| You previously answered '{response_1[i]}'. Can you be more specific and provide the exact species or breed of the animal in the image, if possible? Answer only with the name in lowercase. |

## D  ABLATION STUDY

In the risk feature generation phase, producing risk features for the entire dataset based on risk rules is highly time-intensive. To tackle this issue, we introduce a risk feature mapping approach, wherein the risk features derived from a small labeled dataset $T_s$ in task $T$ are utilized to represent the risk features of a larger unlabeled dataset $T_l$. Consequently, we evaluate the impact of risk features generated with and without risk feature mapping on the performance of risk models.

Table.14 demonstrates that risk features generated through mapping enhance the performance of the risk model when estimating preferences for unlabeled data. The results are contrary to our initial intuition: removing the mapping step actually degraded the performance of the risk model. We analyzed the reasons behind this outcome. The purpose of risk feature mapping is to align the risk features of data from the same distribution, so that the risk model can produce consistent predictions. Without this mapping, the randomness of LLM in risk rules causes the risk features to vary even for data from the same distribution, leading to inaccurate preference estimates. Therefore, using risk feature mapping can make the output of risk model more stable.

Table 14: The impact of various risk feature generation methods on the accuracy of risk model preference estimation.

| Method | RealWorld | MMBench | MMStar |
|---|---|---|---|
| No Mapping | 0.55 | 0.67 | 0.44 |
| Mapping | **0.59** | **0.77** | **0.67** |

## E  DETAILS OF THE EXPERIMENTAL DATA

In this section, we report the detailed data results from the experiment.

### E.1  PREFERENCE PREDICTION ACCURACY

Detailed data on our preference prediction accuracy is presented in Table.15. We set up five distinct seed training risk model. The numbers 1 to 5 in the table correspond to seed values of 666, 888, 999, 1234, and 218 respectively. The "Parameters" column in the table refers to the number of trainable parameters required to fine-tune the target model into a reward model. Our constructed risk model

contains significantly fewer trainable parameters. It is important to clarify that the computation of risk features is not performed during the training of the risk model and therefore is not included in the count of trainable parameters.

Table 15: Details of the data in Table.1

| Data | Method | Parameters | 1 | 2 | 3 | 4 | 5 | Mean±SD |
|------|--------|-----------|------|------|------|------|------|---------|
| Animals | Idefics2-8B | 8B | 43.88 | 46.39 | 46.27 | 45.51 | 45.51 | $45.51 \pm 0.90$ |
| | Llava1.5-7B | 7B | 51.44 | 51.50 | 51.53 | 51.56 | 51.62 | $51.53 \pm 0.06$ |
| | MiniCPM-V 2.6 | 8B | 55.20 | 55.03 | 55.74 | 55.35 | 55.05 | $55.27 \pm 0.26$ |
| | SPRA(Ours) | 7150 | 80.01 | 78.04 | 79.23 | 79.22 | 77.15 | $\mathbf{78.73 \pm 1.01}$ |
| RealWorld | Idefics2-8B | 8B | 61.46 | 55.00 | 56.50 | 55.00 | 53.50 | $56.29 \pm 2.75$ |
| | Llava1.5-7B | 7B | 39.52 | 34.29 | 30.00 | 36.67 | 36.19 | $35.33 \pm 3.15$ |
| | MiniCPM-V 2.6 | 8B | 59.51 | 61.46 | 65.85 | 62.44 | 63.90 | $62.63 \pm 2.15$ |
| | SPRA(Ours) | 7150 | 73.68 | 74.64 | 66.71 | 69.18 | 69.18 | $\mathbf{70.68 \pm 3.00}$ |
| MMbench | Idefics2-8B | 8B | 59.38 | 60.76 | 57.35 | 58.57 | 60.40 | $59.29 \pm 1.24$ |
| | Llava1.5-7B | 7B | 50.86 | 51.33 | 49.60 | 47.63 | 52.02 | $50.29 \pm 1.55$ |
| | MiniCPM-V 2.6 | 8B | 64.82 | 65.59 | 66.82 | 68.05 | 68.82 | $66.82 \pm 1.50$ |
| | SPRA(Ours) | 7150 | 79.68 | 78.93 | 76.21 | 74.90 | 77.43 | $\mathbf{77.43 \pm 1.74}$ |
| MMStar | Idefics2-8B | 8B | 52.76 | 53.24 | 52.28 | 49.40 | 48.68 | $51.27 \pm 1.86$ |
| | Llava1.5-7B | 7B | 45.87 | 46.40 | 44.53 | 43.73 | 44.53 | $45.01 \pm 0.98$ |
| | MiniCPM-V 2.6 | 8B | 56.64 | 54.26 | 50.53 | 53.98 | 55.32 | $54.15 \pm 2.04$ |
| | SPRA(Ours) | 7150 | 69.09 | 65.53 | 66.79 | 64.49 | 68.05 | $\mathbf{66.79 \pm 1.66}$ |
| SeedBench | Idefics2-8B | 8B | 61.43 | 62.00 | 62.03 | 62.06 | 62.63 | $62.03 \pm 0.38$ |
| | Llava1.5-7B | 7B | 44.08 | 44.65 | 44.68 | 44.71 | 45.28 | $44.68 \pm 0.38$ |
| | MiniCPM-V 2.6 | 8B | 56.42 | 56.55 | 56.59 | 56.63 | 56.76 | $56.59 \pm 0.11$ |
| | SPRA(Ours) | 7150 | 71.33 | 69.01 | 70.84 | 68.12 | 70.15 | $\mathbf{69.89 \pm 1.18}$ |
| ScienceQA | Idefics2-8B | 8B | 61.96 | 59.63 | 61.96 | 64.78 | 63.95 | $62.46 \pm 1.79$ |
| | Llava1.5-7B | 7B | 43.19 | 37.99 | 44.02 | 41.22 | 40.14 | $41.31 \pm 2.16$ |
| | MiniCPM-V 2.6 | 8B | 67.60 | 69.66 | 70.22 | 67.23 | 69.10 | $68.76 \pm 1.16$ |
| | SPRA(Ours) | 7150 | 74.79 | 69.96 | 70.57 | 71.76 | 73.77 | $\mathbf{72.17 \pm 1.85}$ |

## E.2 RELATIVE IMPROVEMENT

Detailed data comparing with SCL is presented in Table.16. We set top-p=0.8, top-k=100, and temperature=0.7 in the large model inference configuration. We then ran it five times and recorded the results in the table.

Table 16: Details of the data in Table.3

| Data | Method | 1 | 2 | 3 | 4 | 5 | Mean±SD | Win Rate(%) |
|------|--------|------|------|------|------|------|---------|-------------|
| RealWorldQA | Qwen2-VL-2B | 47.97 | 50.33 | 48.10 | 48.10 | 47.06 | $48.31 \pm 1.08$ | – |
| | **Qwen2-VL-2B+SCL** | 50.64 | 50.20 | 50.39 | 50.92 | 49.80 | $\mathbf{50.39 \pm 0.38}$(↑ 4.31%) | 100 |
| | Qwen2-VL-2B+SPRA | 49.76 | 48.23 | 50.13 | 52.24 | 47.99 | $49.67 \pm 1.53$(↑ 2.81%) | 60 |
| MMbench | Qwen2-VL-2B | 71.63 | 72.54 | 72.24 | 72.20 | 71.88 | $72.10 \pm 0.31$ | – |
| | Qwen2-VL-2B+SCL | 72.15 | 72.30 | 72.55 | 73.15 | 73.10 | $72.65 \pm 0.41$(↑ 0.76%) | 100 |
| | **Qwen2-VL-2B+SPRA** | 74.13 | 73.64 | 74.25 | 74.92 | 73.56 | $\mathbf{74.10 \pm 0.49}$(↑ 2.77%) | 100 |
| MMStar | Qwen2-VL-2B | 38.40 | 38.53 | 37.33 | 37.93 | 37.00 | $37.84 \pm 0.59$ | – |
| | **Qwen2-VL-2B+SCL** | 37.50 | 38.00 | 38.33 | 38.70 | 39.05 | $\mathbf{38.32 \pm 0.54}$(↑ 1.27%) | 80 |
| | Qwen2-VL-2B+SPRA | 38.67 | 38.67 | 38.80 | 37.60 | 37.60 | $38.27 \pm 0.61$(↑ 1.14%) | 80 |
| SeedBench | Qwen2-VL-2B | 60.67 | 60.75 | 60.55 | 60.68 | 61.04 | $60.74 \pm 0.16$ | – |
| | Qwen2-VL-2B+SCL | 61.28 | 61.52 | 60.91 | 61.29 | 62.40 | $61.48 \pm 0.50$(↑ 1.22%) | 100 |
| | **Qwen2-VL-2B+SPRA** | 61.71 | 63.77 | 63.06 | 61.29 | 62.21 | $\mathbf{62.41 \pm 0.90}$(↑ 2.75%) | 100 |
| ScienceQA | Qwen2-VL-2B | 63.01 | 63.16 | 61.53 | 62.62 | 62.82 | $62.63 \pm 0.58$ | – |
| | **Qwen2-VL-2B+SCL** | 65.49 | 64.60 | 63.41 | 64.55 | 62.96 | $\mathbf{64.20 \pm 0.91}$(↑ 2.51%) | 100 |
| | Qwen2-VL-2B+SPRA | 63.70 | 63.81 | 63.71 | 64.25 | 63.76 | $63.84 \pm 0.21$(↑ 1.93%) | 100 |

# F ALGORITHM

In this section, we introduce how risk model is constructed and trained using algorithmic workflows.

The activation matrix is essential for calculating the mean in the algorithm. Therefore, during the inference phase, we employ feature mapping to represent the activation matrix of data outside the subset $D_s$ using the existing activation matrix A. This approach eliminates the need to regenerate A.

---

**Algorithm 1:** Risk model construction & training

---

**Require:** Subset of dataset $D_s = \{d_1, \cdots, d_n\}$, risk rules $R = \{r_1, \cdots, r_m\}$, risk features
$\quad\quad F = \{f_1, \cdots, f_m\}$, activate matrix $A = 0$.
1: **for** $i = 1$ to $n$ **do**
2:    **for** $j = 1$ to $m$ **do**
3:       **if** $d_i$ satisfies $f_j$
4:          $A_{ij} \leftarrow 1$
5:       **else**
6:          $A_{ij} \leftarrow 0$
7:    **end for**
8: **end for**
9: Calculate the mean of the activated samples by column using Equation.2: $\mu = \{\mu_1, \cdots \mu_m\}$.
10: Generate risk labels $y$ for $D_s$.
11: Construct risk model using Equation.6.
12: Optimize risk model using Equation.8.
13: **return** Final risk model.

---

## G    RESOURCE CONSUMPTION OF RISK RULES

In our proposed approach of SPRA, the risk model serves as the reward model. Even though it uses the output of LLMs and VLMs, its tunable parameters are significantly less than the LLMS or VLMs serving as reward models, which usually have billions of parameters.

Our experiments are conducted on two A6000 GPUs. The Table.17 shows that each risk rule requires only a small amount of GPU memory when converted into a risk feature. Every rules are a node in the workflow, they do not occupy GPU memory in parallel. However, if sufficient GPU resources are available, they can be processed in parallel through programming, which would further accelerate the generation of the activation matrix.

Table 17: Risk rule calculation consumption

| Rules | Time (Iter/s) | Memory (GB) |
|---|---|---|
| Statistical | 1.43 | 12 |
| Is animal | 1.84 | 35 |
| Where live | 1.88 | 35 |
| Image feature | 0.30 | 35 |
| Descriptive similarity | 0.20 | 36 |
| Judging the species | 1.35 | 35 |
| Judging the species 2 | 1.30 | 35 |
| Object count and color | 2.00 | 22 |
| Object matching | 2.00 | 22 |

## H    THE USE OF LARGE LANGUAGE MODELS

Our work does not utilize large language models for idea design or writing refinement.

