# OpenReview forum: "Semi-Supervised Preference Learning for Multi-modal Large Models via Risk Analysis"
_ICLR.cc/2026/Conference — Submitted to ICLR 2026_

### Official Review · Reviewer_R8uo · 2025-10-26

**Soundness:** 3
**Presentation:** 2
**Contribution:** 2
**Rating:** 2
**Confidence:** 3

**Summary:**

This paper proposes SPRA, a semi-supervised framework for learning human preferences over multimodal model outputs using limited labeled data. The core idea is to construct a “risk model” that aggregates heuristic risk features (e.g., confidence, consistency, entity matching) into a preference score, which is then used to construct preference pairs for DPO fine-tuning. A CLIP-based mapping mechanism is introduced to scale risk feature generation to unlabeled data.

**Strengths:**

- **Practical Motivation**: Addresses a real bottleneck in preference learning, lack of large-scale human annotations.
- **Systematic Framework**: The pipeline from rule-based risk features, statistical modeling to preference fine-tuning is clearly laid out.
- **Engineering Contribution**: The activation matrix mapping trick using CLIP embeddings is a useful and efficient way to scale pseudo-label generation.
- **Competitive Empirical Results**: Outperforms several strong baselines on preference prediction tasks, despite using far fewer labeled samples.

**Weaknesses:**

- **Conceptual Clarity**: The definition of “risk” is ambiguous and inconsistent across sections. It sometimes refers to uncertainty, sometimes to semantic incorrectness, and sometimes to human preference deviation. The paper conflates “risk” with “preference” without clearly justifying why lower risk implies higher preference.
- **Misleading Claims of "Parameter-Efficiency"**: The paper repeatedly emphasizes that the final risk model is "parameter-efficient" with only "thousands of parameters". This claim is misleading as it conveniently ignores the massive computational overhead of the Risk Feature Generation step. As described in Figures 2 & 3 and Section 3.1.1, generating a single feature vector requires running inference on multiple large "tool" models (e.g., Florence2, Tulu, Qwen2VL-2B), which is also described as "time-consuming" in the paper.
- **Unjustified Methodological Complexity**: The paper introduces a complex theoretical apparatus from financial risk. However, this entire framework appears to be superficial. After deriving the VaR, the authors state it cannot be used for training directly and instead pass it through a Sigmoid function and optimize it using a standard binary cross-entropy loss against "risk labels" of 0 (match) or 1 (mismatch). It is entirely unclear what this convoluted path achieves over a standard, direct binary classification setup (i.e., training the network to directly predict the mismatch probability p). The paper provides no ablation or justification for this added complexity, leaving me unconvinced that it yields any measurable benefit.

**Questions:**

- Could the authors provide a full computational cost analysis (e.g., total FLOPs or GPU-hours) for the entire pipeline, including feature generation, mapping, and risk model training? How does this total cost compare to previous works?
- Can the authors provide an ablation study that directly compares their VaR-based loss function (Eq. 8) against a standard binary cross-entropy loss that directly trains the risk model to predict the risk label using the same features? This is necessary to justify the financial risk framework.
- Can the authors provide a formal and concise definition of "risk" mentioned in this paper?

I am willing to raise my rating if the authors can address my points above.

---

> ### Author Response · Authors · 2025-11-25
> **Response to Reviewer  R8uo; Question1-2**
>
> **Question 1**:	Conceptual Clarity: The definition of “risk” is ambiguous and inconsistent across sections. It sometimes refers to uncertainty, sometimes to semantic incorrectness, and sometimes to human preference deviation. The paper conflates “risk” with “preference” without clearly justifying why lower risk implies higher preference.
>
> **Response 1**: Thanks for the comment, We have provided a detailed explanation of what risk at Line 66 of the paper: the proposed approach generates multiple outputs on each unlabeled data, which are then ranked by their deviation risk. Lower-risk results are prioritized as preferred outputs (labels). As mentioned in line 180, model outputs that align with human preferences are considered low-risk. This is consistent with human logic.
>
> **Question 2**: Misleading Claims of "Parameter-Efficiency": The paper repeatedly emphasizes that the final risk model is "parameter-efficient" with only "thousands of parameters". This claim is misleading as it conveniently ignores the massive computational overhead of the Risk Feature Generation step. As described in Figures 2 & 3 and Section 3.1.1, generating a single feature vector requires running inference on multiple large "tool" models (e.g., Florence2, Tulu, Qwen2VL-2B), which is also described as "time-consuming" in the paper.
>
> **Response 2**: Thank you for the raised concern. When this paper talks about “parameter-efficiency”, we mean the size of reward models that can be fine-tuned. In our proposed approach of SPRA, the risk model serves as the reward model. Even though it uses the output of LLMs and VLMs, its tunable parameters are significantly less than the LLMS or VLMs serving as reward models, which usually have billions of parameters. We will improve presentation to clarify this claim and address the concern.
>
> It is noteworthy that the tool models you mentioned are not involved in training; they only produce inference results. Using the prompts provided in Appendix C.2, you'll find their outputs are quite brief, so they won't consume much time either. These results are used to build risk models, which are far easier and less resource-intensive to train than a large model. The entire process can be completed on just two A6000 GPUs. However, this setup is far from sufficient for training a large model, especially without a fully annotated preference dataset. Algorithm 1 in Appendix F has already illustrated this process:
>
> **Require**:
> - Subset of dataset \(D_s = \{d_1, \dots, d_n\}\)
> - Risk rules \(R = \{r_1, \dots, r_m\}\)
> - Risk features \(F = \{f_1, \dots, f_m\}\)
> - Activation matrix \(A = 0\)
>
> ---
>
> 1. **For** \(i = 1\) to \(n\):
>     1. **For** \(j = 1\) to \(m\):
>         - **If** sample \(d_i\) satisfies feature \(f_j\):
>           → Set \(A_{ij} = 1\)
>         - **Else**
>           → Set \(A_{ij} = 0\)
>     2. **End for**
> 2. **End for**
>
> 3. Compute column-wise mean of activated samples using Eq. (2):
>    → \(\mu = \{\mu_1, \dots, \mu_m\}\)
>
> 4. Generate risk labels \(y\) for \(D_s\).
>
> 5. Construct risk model using Eq. (6).
>
> 6. Optimize risk model using Eq. (8).
>
> 7. **Return** final trained risk model.

---

> > ### Author Response · Authors · 2025-12-02
> > **Revision based on Question1-2**
> >
> > **Q1**.Conceptual Clarity: The definition of “risk” is ambiguous and inconsistent across sections. It sometimes refers to uncertainty, sometimes to semantic incorrectness, and sometimes to human preference deviation. The paper conflates “risk” with “preference” without clearly justifying why lower risk implies higher preference.
> >
> > **Revision**: 1) We clarify at line 64 regarding the relationship between risk and preference:
> >
> >     “It generally quantifies the relationship between a model's outputs and human preferences by a risk model. The higher the risk, the greater the deviation of an output from human preferences and vice versa.”
> >
> > 2). we emphasize the advantage of risk over traditional single value measure at line 115:
> >
> >     “It measures risk, or preference alignment, by a distribution. Compared with single values
> > predicted by existing reward models, distributions can effectively capture fluctuation risk,
> > whose necessity has been widely accepted by existing risk research;”
> >
> > **Q2**.Misleading Claims of "Parameter-Efficiency": The paper repeatedly emphasizes that the final risk model is "parameter-efficient" with only "thousands of parameters". This claim is misleading as it conveniently ignores the massive computational overhead of the Risk Feature Generation step. As described in Figures 2 & 3 and Section 3.1.1, generating a single feature vector requires running inference on multiple large "tool" models (e.g., Florence2, Tulu, Qwen2VL-2B), which is also described as "time-consuming" in the paper.
> >
> > **Revision**: 1). Our major contribution is on automatically constructing high-quality preference
> > data with only a limited label information, effectively reducing the cost of manual annotation. We highlighted our contributions in line 107. The aspect of parameter efficiency is not a central focus of this paper; rather, it is included only as a minimal supporting component:
> >
> >      “It incorporates human-centric priors with a small-sized risk model, making the process of preference finetuning more interpretable and resource-efficient. Compared with existing single reward models, which are usually prone to introducing bias, the proposed risk model can effectively reduce bias by fusing independent outputs from various large models.”
> >
> > 2). By parameter efficiency, we mean the size of reward models that can be fine-tuned. We move the results on model size to Appendix G:
> >
> >      “In our proposed approach of SPRA, the risk model serves as the reward model. Even though it uses the output of LLMs and VLMs, its trainable parameters are significantly less than the LLMS or VLMs serving as reward models, which usually have billions of parameters.”
> >
> > 3). We add the resource consumption caused by risk-feature computation from the runtime logs to the appendix, in Section G, and marked it in red:
> >
> >      "Our experiments are conducted on two A6000 GPUs. The Table.18 shows that each risk rule requires only a small amount of GPU memory when converted into a risk feature. Every rules are a node in the workflow, they do not occupy GPU memory in parallel. However, if sufficient GPU resources are available, they can be processed in parallel through programming, which would further accelerate the generation of the activation matrix."

---

> ### Author Response · Authors · 2025-11-25
> **Response to Reviewer  R8uo; Question3**
>
> **Question 3**: Unjustified Methodological Complexity: The paper introduces a complex theoretical apparatus from financial risk. However, this entire framework appears to be superficial. After deriving the VaR, the authors state it cannot be used for training directly and instead pass it through a Sigmoid function and optimize it using a standard binary cross-entropy loss against "risk labels" of 0 (match) or 1 (mismatch). It is entirely unclear what this convoluted path achieves over a standard, direct binary classification setup (i.e., training the network to directly predict the mismatch probability p). The paper provides no ablation or justification for this added complexity, leaving me unconvinced that it yields any measurable benefit.
>
>
> **Response 3**: The crucial benefit of risk model over a simple binary classifier is that it measures risk by a distribution, which captures fluctuation risk. The necessity of fluctuation risk is inspired by extensive risk research in investment theory. Therefore, the proposed risk model leverages a normal distribution to represent match/mismatch probability. All of our technical designs are centered around how to accurately estimate the normal distribution. Therefore, the methodological complexity is justified.
>
> Theoretically, the reason samples follow normal distribution is that individual samples (input, output) conform to a Bernoulli distribution: for verifiable tasks, the output is either correct or incorrect. The Bernoulli distribution of multiple samples can be approximated as a normal distribution. By measuring the risk value of a model's output at a 90% confidence level, we can assess that we have 90% confidence that the risk value is a specific value.
>
> The calculated VaR serves as a measure of risk. This number cannot be directly predicted and is not confined to the range of 0 to 1; it can be either very large or very small. Therefore, we use a sigmoid function to constrain it between 0 and 1, which similarly represents the scale of risk while being easier to optimize. Cross-entropy is not merely a classifier loss function; it primarily represents an expectation. Our optimization goal is to maximize this expectation—that is, to maximize the risk value (the maximum expected risk per sample). We select cross-entropy to optimize the model, not because we view it as a binary classification model. Mathematically, cross-entropy is the most appropriate choice for this purpose.

---

> > ### Comment · Reviewer_R8uo · 2025-11-25
> >
> > Thank you to the authors for their rebuttal. However, my major concerns have not been addressed:
> >
> > Computational Cost: The rebuttal's claim that tool models are "not involved in training" is incorrect. To calculate the loss function (Eq. 8) during training, which requires to run inference on multiple large tool models (e.g., Qwen2VL, Tulu) for every sample. Thus, the heavy inference overhead is an unavoidable prerequisite for the training process. The authors have not provided any analysis on computation efficiency compared with other baselines.
> >
> > Missing Empirical Evidence: The rebuttal failed to provide the requested ablation study (VaR Loss vs. Standard BCE). Theoretical explanations alone cannot justify the method's complexity without empirical proof in this field.
> >
> > Therefore, I maintain my Reject rating.

---

> > > ### Author Response · Authors · 2025-11-25
> > >
> > > We would like to reiterate that the main focus of our work is not model optimization. Cross entropy is used to maximize the expected value of our model. What is VaR loss? We did not mention it in our paper.
> > >
> > >
> > > The generation of risk rules is independent of model training, so the inference of large models does not affect the fine-tuning process of large models. In other words, the generated activation matrices are saved in advance and are only a few KB matrix files that can be loaded when in use. The algorithm flowchart we provided is already very clear about this point. GRPO is an excellent reinforcement learning method. In the process of fine-tuning the large model, the generation of the large model and the training of the reward model were also used, but it is still widely used.

---

> > > > ### Comment · Reviewer_R8uo · 2025-11-25
> > > >
> > > > Thank you for the further clarifications. However, I still have reservations regarding the empirical justification and the practical scalability of the proposed framework.
> > > >
> > > > 1. The "VaR loss," I am referring to the objective function defined in Equation 8, which explicitly optimizes the Value-at-Risk calculated in Equation 7. My concern is not about the terminology, but about the empirical necessity of this complex design. Theoretical motivation from investment theory is interesting, but for a machine learning contribution, we need to know if this complexity yields tangible benefits. Specifically, does the proposed VaR-based objective provide a measurable performance improvement over a simpler baseline, such as a standard binary classifier?
> > > >
> > > > 2. I understand that the risk rule generation is independent of the final model training and can be pre-computed. However, the time cost of this pre-computation is still a critical part of the total pipeline. If generating the activation matrix requires running inference on multiple large tool models (VLM + LLM) for every data sample, this "offline" cost may cannot be ignored. As noted in Line 358 of the paper, this process "may be very time-consuming." Therefore, I would like to ask authors to provide more details about this part.
> > > >
> > > > I am still willing to raise my score if the two concerns above can be adequately addressed or clarified with evidence.

---

> > ### Author Response · Authors · 2025-12-02
> > **Revision based on Q3**
> >
> > **Q3**. Unjustified Methodological Complexity: The paper introduces a complex theoretical apparatus from financial risk. However, this entire framework appears to be superficial. After deriving the VaR, the authors state it cannot be used for training directly and instead pass it through a Sigmoid function and optimize it using a standard binary cross-entropy loss against "risk labels" of 0 (match) or 1 (mismatch). It is entirely unclear what this convoluted path achieves over a standard, direct binary classification setup (i.e., training the network to directly predict the mismatch probability p). The paper provides no ablation or justification for this added complexity, leaving me unconvinced that it yields any measurable benefit.
> >
> >
> > **Revision**: 1) We refined the contributions of the paper and clarified in line 115 that SPRA is not a simple classification task, but rather learns a distribution that determines whether the model’s output carries risk:
> >
> >     “It measures risk, or preference alignment, by a distribution. Compared with single values predicted by existing reward models, distributions can effectively capture fluctuation risk, whose necessity has been widely accepted by existing risk research”
> >
> > 2). In line 180, we describe the modeling process of the risk model, clarifying that it is formulated as distribution modeling rather than classification:
> >
> >      As usual, SPRA treats the matching probability of each data pair as a random variable that follows a Bernoulli distribution with parameter p, where p follows a Beta(α, β) distribution. The α and β are the shape parameters of the Beta distribution. From a statistical perspective, the Beta distribution can be approximated by a normal distribution when α + β ≥ 10. Specifically, in the context of the task scenario, α + β represents the total number of samples.
> >
> > 3). We highlight line 344 to indicate that VaR is a quantile of a distribution and therefore does not have a fixed boundary; we use a sigmoid function to map it into the range of 0 to 1. Therefore, we quantify VaR using a value within the closed interval [0,1]. Its meaning is described in line 353 of the manuscript:
> >
> >     “Specifically, the risk label is recorded as 0 when the output of model matches the ground truth and as 1 when it differs.”

---

> ### Author Response · Authors · 2025-11-26
>
> **R1**: We compared multiple large models in Table 1, and the Table 8 in the appendix provides the prompt used by these large models for output. You can see that they actually play the role of classifiers, classifying whether one of the responses of the target model match human preferences. This is a common method of preference evaluation: evaluating the output preferences of the target model through higher-order large models. At this point, it can be understood as a classifier.
>
> What we hope you know is that the risk value is a quantile of the distribution, which we do not know and cannot determine. We cannot quantify what is large or small because there is no fixed boundary value.  Therefore, we map it between 0-1 using sigmoid for optimization, which not only ensures boundary properties but also finds suitable methods for optimization.
>
>
> **R2**: The “time-consuming”  we refer to comes from the full sentence: “it becomes necessary to input risk features for the unlabeled portion of the data. The risk feature generation process may be very time-consuming, especially when dealing with large datasets.” This describes the process of generating risk features for the large amount of remaining unlabeled data. Because the unlabeled part is substantial, producing risk features step by step—following the procedure described above—requires considerable time. To address this, we propose a feature-mapping strategy, which uses known risk features to represent unknown ones, thereby greatly reducing the overall computation time.
>
> We have analyzed the log files of the experimental runs, and summarized the runtime and GPU memory consumption for each individual risk rule.
>
> | Rules          | Time (iter/s) | Memory (GB) |
> |----------------|----------------|-------------|
> | Statistical    | 1.43           | 12         |
> | Is animal      | 1.84           | 35         |
> | Where live     | 1.88           | 35         |
> | Image feature     | 0.30           | 35         |
> | Descriptive similarity    | 0.20           | 36         |
> | Judging the species     | 1.35           | 35         |
> | Judging the species 2    | 1.30           | 35         |
> | Object count and color   | 2.00           | 22         |
> | Object matching      | 2.00           | 22         |
>
> The table shows that the memory usage of each rule is minimal. This is far more memory- and time-efficient than training a large model. Every rules are a node in the workflow, they do not occupy GPU memory in parallel. However, if sufficient GPU resources are available, they can be processed in parallel through programming, which would further accelerate the generation of the activation matrix.

---

### Official Review · Reviewer_bPAR · 2025-10-30

**Soundness:** 2
**Presentation:** 2
**Contribution:** 2
**Rating:** 4
**Confidence:** 3

**Summary:**

The paper proposes a new risk-based preference modeling approach that enables model improvement with limited labeled data.
Here, the risk is defined using features derived from image-understanding questions (feature extraction questions). It is measured by how much the model’s answer changes when such features are included. A risk model is trained to approximate this risk using a Gaussian distribution, and the main model is then refined via DPO training on data labeled by the risk model, using self-corrected answer pairs generated by the model itself.

**Strengths:**

1. The concept of extracting additional features used in human reasoning and leveraging them for risk computation is novel. Although the paper employs a limited rule-based approach for feature extraction, this idea has the potential to be extended toward a broader and more comprehensive analysis.

2. The evaluations conducted on diverse datasets such as Animals, RealWorldQA, and MMBench demonstrate that the proposed method is not limited to specific scenarios, but can be widely applied across various contexts.

3. It is impressive that the risk model, despite its small size (only 7K parameters), achieves preference classification performance comparable to vision-language models (VLMs) that are over a thousand times larger.

**Weaknesses:**

1. The performance improvements reported in the paper appear marginal. For instance, in the case of Qwen2-VL-2B, the improvement is only around 1–2%, which is roughly within the range of statistical variance. It is therefore unclear whether these gains are statistically significant.
2. The paper compares its method to SCL trained on the full dataset, but it should also include a comparison with SCL trained on the same subset of data used by SPRA. Since SPRA achieves comparable performance while using only about 40% of the labels on most benchmarks, its apparent data efficiency advantage might not be as strong as claimed. A difference of around 40% may not significantly impact the overall training performance.
3. It is uncertain how effective this approach would remain when applied to more advanced models. For example, with recent models such as Qwen2.5 or 3 VL (8B-scale or even small), even simple self-correction or self-rewarding mechanisms may yield similar improvements—especially given that the performance gains reported in this paper are quite small. A fair evaluation should therefore include comparisons with these alternative, simpler methods, as well as with "reward model'' approaches, since the proposed risk model plays a similar role. The absence of such comparative analyses makes it difficult to fully assess the validity and relative effectiveness of the proposed method.

**Questions:**

1. Are there any additional baselines beyond SCL? For example, would it be possible to include a simple SFT baseline for comparison?

2. The process of extracting risk features is based on heuristically defined rules and involves generation through a model. Since these predefined rules cannot fully capture the model’s overall feature and are heuristically chosen, they may introduce bias. Furthermore, the model used to generate problem instances could inherently contain bias. Did you observe any such potential biases arising from either the rule definitions or the problem generation itself?

---

> ### Author Response · Authors · 2025-11-25
> **Response to Reviewer bPAR; Question1**
>
> **Question1**: The performance improvements reported in the paper appear marginal. For instance, in the case of Qwen2-VL-2B, the improvement is only around 1–2%, which is roughly within the range of statistical variance. It is therefore unclear whether these gains are statistically significant.
>
> **Response1**: Thanks for your review. For preference learning, 1-3% improvements can be generally considered substantial for VLM downstream task, e.g., question-answering tasks[1, 2, 3]. We have listed below the improvements achieved by SCL based on DPO on question-answering datasets, demonstrating that the enhancements of the model in our paper also fall within the effective range. All the reported results are the averages over five runs. The variances among different runs are actually very small, <0.6%. Therefore, the reported improvements are statistically significance. It is worthy to emphasize that our primary contribution is on constructing a preference dataset approximating ground-truth using only a limited amount of labeled data. Our proposed approach of SPRA can significantly reduce manual annotation cost while delivering highly competitive quality compared with the approach using all the ground-truths.
>
>
> | Model                     | RealWorldQA | MMStar | MMBench | SEEDBench | ScienceQA | MMT-Bench | MMMU  | AI2D  | Rank |
> |---------------------------|-------------|--------|---------|-----------|-----------|-----------|--------|--------|------|
> | **LLaVA-V1.5-13B**        |             |        |         |           |           |           |        |        |      |
> | +SFT                     | **56.59**   | 36.33  | 75.16   | **69.32** | 71.88     | 40.08     | 34.88 | 55.95 | 2.75 |
> | +SCL               | 56.03       | **37.60** | **75.40** | 68.56      | **72.16** | **41.16** | **35.87** | **58.92** | **1.38** |
> | **InternLM-XComposer-2-7B** |          |        |         |           |           |           |        |        |      |
> | +SFT                     | 62.03       | 48.90  | **77.80** | **70.20** | 79.50     | **50.20** | 57.95 | 80.66 | 1.75 |
> | +SCL              | **62.23**   | **49.60** | 77.00   | 70.60      | **79.90** | **50.20** | **58.20** | **81.11** | **1.13** |
>
>
>
> The experimental results presented in this paper were obtained from five independent runs. Detailed experimental results, showing that most variances are small(<0.6%). Compared to other methods, it exhibits considerable variance. Due to character limitations, we have listed only a portion of the results here.
> | Data         | Method             | 1      | 2      | 3      | 4      | 5      | Mean±SD                                      | Win Rate (%) |
> |--------------|--------------------|--------|--------|--------|--------|--------|------------------------------------------------|--------------|
> | **RealWorldQA** | Qwen2-VL-2B        | 47.97  | 50.33  | 48.10  | 48.10  | 47.06  | 48.31 ± 1.08                                  | –            |
> |              | Qwen2-VL-2B+SCL    | 50.64  | 50.20  | 50.39  | 50.92  | 49.80  | **50.39 ± 0.38 (↑ 4.31%)** | **100**      |
> |              | Qwen2-VL-2B+SPRA   | 49.76  | 48.23  | 50.13  | 52.24  | 47.99  | 49.67 ± 1.53 (↑ 2.81%) | 60           |
> |--------------|--------------------|--------|--------|--------|--------|--------|------------------------------------------------|--------------|
> | **MMbench**  | Qwen2-VL-2B        | 71.63  | 72.20  | 72.20  | 71.88  | 72.20  | 72.10 ± 0.31                                  | –            |
> |              | Qwen2-VL-2B+SCL    | 72.15  | 72.30  | 72.55  | 73.15  | 73.10  | 72.65 ± 0.41 (↑ 0.76%) | **100**      |
> |              | Qwen2-VL-2B+SPRA   | 74.13  | 73.43  | 73.74  | 73.56  | 73.56  | **74.10 ± 0.49 (↑ 2.77%)** | **100**      |
> |--------------|--------------------|--------|--------|--------|--------|--------|------------------------------------------------|--------------|
> | **MMStar**   | Qwen2-VL-2B        | 38.40  | 38.53  | 37.33  | 37.93  | 37.00  | 37.84 ± 0.59                                  | –            |
> |              | Qwen2-VL-2B+SCL    | 37.50  | 38.00  | 38.33  | 38.70  | 39.05  | **38.32 ± 0.54 (↑ 1.27%)** | **80**           |
> |              | Qwen2-VL-2B+SPRA   | 38.67  | 37.67  | 38.87  | 39.00  | 37.33  | 38.27 ± 0.61 (↑ 1.14%) | 60           |
> |--------------|--------------------|--------|--------|--------|--------|--------|------------------------------------------------|--------------|
>
>
>
> More detailed experimental results are presented in Appendix E, indicating that our findings are not coincidental.
>
> [1]. SimPO: Simple Preference Optimization with a Reference-Free Reward. NeurIPS (2024)
>
> [2]. β-DPO: Direct Preference Optimization with Dynamic β. NeurIPS (2024)
>
> [3]. Self-Correction is More than Refinement: A Learning Framework for Visual and Language Reasoning Tasks. ACL Findings (2025)

---

> > ### Author Response · Authors · 2025-12-02
> > **Revision based on Question1**
> >
> > **Q1**.The performance improvements reported in the paper appear marginal. For instance, in the case of Qwen2-VL-2B, the improvement is only around 1–2%, which is roughly within the range of statistical variance. It is therefore unclear whether these gains are statistically significant.
> >
> > **Revision**: 1) . We add the following description in Section 4 to make the experimental logic clearer and emphasize that the primary goal of SPRA is preference prediction rather than model fine-tuning:
> >
> >     This section evaluates the efficacy of the proposed SPRA by an empirical study on real benchmark datasets. Since the major contribution of SPRA is using a small amount of labeled data to automatically annotate a large volume of unlabeled data, which can then be leveraged to fine-tune preferences of VLMs. Therefore, our experiments have two primary objectives: 1) to demonstrate that the preference labels generated by SPRA are considerably more accurate than the existing alternatives of large reward models; 2) to demonstrate that using only a limited amount of ground-truth labels, SPRA can achieve highly competitive preference learning compared to the classical preference fine-tuning method of SCL using all the ground-truths.
> >
> > 2). we added new references at line 498 showing that fine-tuned models—whether trained on SPRA-labeled or on ground-truth–labeled preference datasets, both results are shown as reliable on verifiable tasks:
> >
> >     It is worth noting that our experiments only applied DPO to the model without performing supervised fine-tuning(SFT), in order to align with the experimental results in SCL. With only preference learning, pretrained large models are inherently difficult to improve significantly. For instance, inverifiable tasks (e.g., Q&A), leading optimization methods like SimPO(Meng et al., 2024) and β-DPO(Wu et al., 2024) have demonstrated that an improvement of nearly 3% is typically considered effective.
> >
> >
> > 3). We add an experiment demonstrating the performance improvement obtained by VLMs on a dataset constructed using the same 40% ground-truth data as SPRA. The specific description can be found in lines 513:
> >
> >     “As mentioned above, we have implemented two versions of SCL, SCL-100\%, which uses all the ground-truths in the training data of benchmarks, and SCL-40\%, which uses the same amount of labeled data as SPRA.”
> > The related description is also at line 522:
> >
> >     Secondly, it can be observed that if using only 40\% of the ground-truth in the training dataset, the performance of SCL is consistently worse than SPRA. This suggests that leveraging a small set of labeled examples to annotate a large amount of unlabeled data is indeed meaningful, as it leads to more substantial performance gains.

---

> ### Author Response · Authors · 2025-11-25
> **Response to Reviewer bPAR; Question2**
>
> **Question 2**: The paper compares its method to SCL trained on the full dataset, but it should also include a comparison with SCL trained on the same subset of data used by SPRA. Since SPRA achieves comparable performance while using only about 40% of the labels on most benchmarks, its apparent data efficiency advantage might not be as strong as claimed. A difference of around 40% may not significantly impact the overall training performance.
>
> **Response2**: Thanks for the insightful comment. In the original SCL paper, they fine-tuned the large model using 100% of the training data. To ensure experimental alignment with their approach, we also used the SCL of 100% data as a comparison scheme to highlight the validity of the dataset constructed by SPRA.
>
> Our proposed approach of SPRA involves using a small amount of labeled data to automatically annotate a large volume of unlabeled data in a manner that achieves the same quality as human-labeled ground truth. Firstly, we demonstrate that the preference labels generated by SPRA are considerably more accurate than the existing alternatives of large models. Secondly, we demonstrate that using only a limited amount of ground-truth labels, SPRA can achieve highly competitive preference learning compared to the SCL using all the ground-truths.
>
> Based on your suggestion, we have also tested Qwen2's performance after training on 40% of the data. The detailed evaluation results, which will be include in the appendix, are as follows:
>
> | Method   | realworld | mmstar | mmbench | seedbench | science |
> |----------|-----------|--------|---------|-----------|---------|
> | baseline | 48.31     | 37.84  | 72.10   | 60.74     | 62.63   |
> | SCL-40%  | 48.37     | 36.93  | 72.12   | 60.94     | 62.95   |
> | SCL      | 50.39     | 38.32  | 72.65   | 61.48     | 64.20   |
> | SPRA     | 49.67     | 38.27  | 74.10   | 62.41     | 63.84   |
>
> It can be observed that if using only 40% of the ground-truths in the training dataset, the performance of SCL is consistently worse than SPRA, with the margins between 1-3%, which are comfortably above the variances (<0.6%).

---

> > ### Author Response · Authors · 2025-12-02
> > **Revision based on Question2-3**
> >
> > **Q2**.The paper compares its method to SCL trained on the full dataset, but it should also include a comparison with SCL trained on the same subset of data used by SPRA. Since SPRA achieves comparable performance while using only about 40% of the labels on most benchmarks, its apparent data efficiency advantage might not be as strong as claimed. A difference of around 40% may not significantly impact the overall training performance.
> >
> > **Revision**: In Table 3, we add the 40% improvement in model performance. The revision here is consistent with that in Response 1.
> >
> > **Q3**.It is uncertain how effective this approach would remain when applied to more advanced models. For example, with recent models such as Qwen2.5 or 3 VL (8B-scale or even small), even simple self-correction or self-rewarding mechanisms may yield similar improvements—especially given that the performance gains reported in this paper are quite small. A fair evaluation should therefore include comparisons with these alternative, simpler methods, as well as with "reward model'' approaches, since the proposed risk model plays a similar role. The absence of such comparative analyses makes it difficult to fully assess the validity and relative effectiveness of the proposed method.
> >
> > **Revision**: 1). We added new references in the related work section at Line 138 to discuss prior studies on self-correction and self-rewarding:
> >
> >     “Although data-free self-correction and self-rewarding methods are aligned with our goals, they have been shown to be fundamentally limited (Huang et al., 2024; Kumar et al., 2024; Wu et al., 2024b; Kamoi et al., 2024). These works indicate that large-scale fine-tuning is key to achieving stable self-correction or rewarding. Because without extra fine-tuning, a model’s abilities stay the same.”
> >
> > 2). In line 480 of the manuscript, we clarified that the experiments are conducted on both small and large models:
> >
> >     “To evaluate the efficacy of SPRA for preference fine-tuning, we fine-tune both small and large benchmark models, the small Qwen2-VL-2B and the relatively larger Idefics3-8B”

---

> ### Author Response · Authors · 2025-11-25
> **Response to Reviewer bPAR; Question3**
>
> **Questopn3**.	It is uncertain how effective this approach would remain when applied to more advanced models. For example, with recent models such as Qwen2.5 or 3 VL (8B-scale or even small), even simple self-correction or self-rewarding mechanisms may yield similar improvements—especially given that the performance gains reported in this paper are quite small. A fair evaluation should therefore include comparisons with these alternative, simpler methods, as well as with "reward model'' approaches, since the proposed risk model plays a similar role. The absence of such comparative analyses makes it difficult to fully assess the validity and relative effectiveness of the proposed method.
>
> **Response3**: Thank you for the insightful comment. As far as we know, the performance of self-correction is not stable; preference optimization is still essential for adapting a model to downstream tasks. For instance, the recent work of [1] indicates that simple self-correction is unreliable in most cases and may even lead to performance degradation. Similarly, the work of [2] demonstrates that even when training models using self-correcting SFT, the models still fail to provide useful self-correction capabilities.
>
> Furthermore, the efficacy of our proposed approach has solid theoretical grounding,  because in the setting of preference learning, higher-quality preference labeled data would mean higher performance for LLM or VLMs. Our proposed approach can automatically construct high-quality preference data with only limited label information, effectively reducing the cost of manual annotation. Therefore, we compare it with the advanced models in terms of preference accuracy. e.g., Idefics2-8B, Llava1.5-7B, MiniCPM-V 2.6 -8B, which similarly serve as reward models. Our evaluation results as shown in Table 1 demonstrate that it performs considerably better than these advanced models.
>
> It is noteworthy that we have already used both small and large models, e.g., Qwen2.5-2B and Idefics3-8B, in the evaluation of preference fine-tuning.  The detailed results have been presented in Table 2 of the paper as follows:
> | Models           | Animals (%)      | RealWorldQA (%)    | MMBench (%)      | MMStar (%)        | SeedBench (%)     | ScienceQA (%)     |
> |------------------|------------------|---------------------|-------------------|--------------------|--------------------|--------------------|
> | Qwen2-VL-2B       | 68.24 (±0.08)    | 48.31 (±1.21)       | 72.10 (±0.35)     | 37.84 (±0.66)      | 60.74 (±0.18)      | 62.63 (±0.65)      |
> | **Qwen2-VL-2B+ft** | **69.92 (±1.61)** | **49.67 (±1.53)**   | **74.10 (±0.49)** | **38.27 (±0.61)**  | **62.41 (±0.90)**  | **63.84 (±0.21)**  |
> | Idefics3-8B       | 73.17 (±1.20)    | 60.39 (±1.08)       | 79.87 (±2.15)     | 43.67 (±1.17)      | 66.81 (±2.30)      | 85.47 (±1.21)      |
> | **Idefics3-8B+ft** | **74.27 (±1.11)** | **60.91 (±1.15)**   | **79.96 (±1.16)** | **43.73 (±1.04)**  | **66.90 (±1.11)**  | **85.91 (±0.98)**  |
>
> It is noteworthy that achieving significant improvements on verifiable tasks for models with large parameters is challenging. For instance, we refer to the improvement of the SOTA method SimPO over DPO (at the 7B scale), indicating that the gains from SPRA-constructed datasets for models are within the expected range. The following results are from SimPO[3]:
>
> | Method | MMLU (5) | ARC (25) | HellaSwag (10) | TruthfulQA (0) | Winograd (5) | GSM8K (5) | Average |
> |--------|----------|-----------|----------------|----------------|--------------|-----------|---------|
> | DPO    | 58.48    | 61.26     | 83.59          | 53.06          | 76.80        | 21.76     | 59.16   |
> | SimPO  | 59.21    | 62.63     | 83.60          | 50.68          | 77.27        | 22.21     | 59.27   |
>
>
> Based on your comments, we will strengthen the paper by using more advanced models for preference fine-tuning.
>
> [1]. Large Language Models Cannot Self-Correct Reasoning Yet, ICLR(2024).
>
> [2]. Training Language Models to Self Correct via Reinforcement Learning ICLR(2025).
>
> [3]. SimPO: Simple Preference Optimization with a Reference-Free Reward. NeurIPS (2024)

---

> ### Author Response · Authors · 2025-11-25
> **Response to Reviewer bPAR; Question4-5**
>
> **Question 4**: Are there any additional baselines beyond SCL? For example, would it be possible to include a simple SFT baseline for comparison?
>
> **Response4**: Thank you for your suggestion. Table 1 We have compared our model with numerous state-of-the-art reward models. We selected SCL due to its utilisation of ground truth, thereby enabling verification of whether the preference dataset constructed by SPRA can rival ground-truth.
>
> SFT and preference learning are not the same fine-tuning approach. SFT is a supervised fine-tuning approach. This method cannot learn the gap between preference and non-preference. (rejected). Since our task targets preference learning, using DPO is the mainstream solution. Finally, it is noteworthy to emphasize that our primary contribution is on constructing a preference dataset closely aligned with ground truth using only a limited amount of labeled data, but not on preference fine-tuning methods. Therefore, the work on DPO (used by SCL and our proposed SPRA) and SFT are orthogonal to ours.
>
>
>
> **Question 5**.  The process of extracting risk features is based on heuristically defined rules and involves generation through a model. Since these predefined rules cannot fully capture the model’s overall feature and are heuristically chosen, they may introduce bias. Furthermore, the model used to generate problem instances could inherently contain bias. Did you observe any such potential biases arising from either the rule definitions or the problem generation itself?
>
> **Response 5**: Thanks for the insightful comment. Current single-reward models are more prone to introducing bias. Our risk model effectively reduces bias by fusing outputs from different large models. In terms of experiments, we repeated each experiment five times. The results alleviated concerns about the introduction of errors in risk rules. In fact, we fitted a normal distribution, and the mean of this distribution was derived from risk rule assessments. When the sample size is large, partial errors do not significantly impact the overall mean calculation, as this aligns with the Law of Large Numbers.

---

> > ### Author Response · Authors · 2025-12-02
> > **Revision based on Question4-5**
> >
> > **Q4**.Are there any additional baselines beyond SCL? For example, would it be possible to include a simple SFT baseline for comparison?
> >
> > **Revision**: 1). At line 473, we introduce SCL as a preference-based optimization paradigm that is independent of SFT:
> >
> > 	“In this section, following the approach of SCL proposed by He et al., we construct a preference dataset and then fine-tune the model using Direct Preference Optimization (DPO). However, unlike the original SCL, which relies on ground-truth labels to determine whether a model output aligns with human preferences, our SPRA leverages the risk assessment provided by the risk model.”
> >
> > 2). At line 495, we highlight why we did not use SFT, as shown below:
> >
> >     “It is worth noting that our experiments only applied DPO to the model without performing supervised fine-tuning(SFT), in order to align with the experimental results in SCL. With only preference learning, pretrained large models are inherently difficult to improve significantly.”
> >
> > **Q5**.The process of extracting risk features is based on heuristically defined rules and involves generation through a model. Since these predefined rules cannot fully capture the model’s overall feature and are heuristically chosen, they may introduce bias. Furthermore, the model used to generate problem instances could inherently contain bias. Did you observe any such potential biases arising from either the rule definitions or the problem generation itself?
> >
> > **Revision**: In line 279, we add a discussion of these risk features to clarify the stability of the SPRA approach:
> >
> >     “A risk feature is supposed to indicate the uncertainty of an output from one perspective. However, using any individual risk feature to describe the risk inherent in a large model's output is usually inadequate. On the other hand, any single risk feature (or reward model) would be prone to introducing bias. Therefore, our proposed approach leverages various risk features, which are supposed to be constructed based on various outputs from different large models. It can be expected that fusing outputs from different large models would effectively reduce preference bias.”

---

### Official Review · Reviewer_j9V1 · 2025-10-31

**Soundness:** 2
**Presentation:** 2
**Contribution:** 2
**Rating:** 4
**Confidence:** 4

**Summary:**

This paper propose a Semi-supervised Preference learning approach based on Risk Analysis（SPRA），which can measures preference by a risk model. Experimental results show that this method achieves good performance, while significantly reducing computational overhead and simplifying reward optimization. However, the method proposed in this paper does not seem to be particularly innovative. Moreover, the experimental hyperparameters and implementation details are not sufficiently described or analyzed, making it difficult to understand what the model has actually learned regarding risk handling.

**Strengths:**

1. This paper presents a seemingly effective approach for handling risks and validates it through experiments.
2. The authors' writing in the methodology section is relatively clear.

**Weaknesses:**

1. My main concern is that the proposed method in this paper appears more like a technical implementation strategy, lacking sufficient theoretical support. Although some experimental results are provided, they are not comprehensive enough. I recommend adding experiments under different hyperparameters, such as varying temperatures and learning rates, to better validate the method's effectiveness and robustness.
2. The paper compares the proposed SPRA method with SCL in experiments, and the results show that SPRA does not demonstrate a significant performance advantage. Although the authors mention that SPRA uses only a small amount of ground-truth labels compared to SCL, this advantage should primarily be attributed to the inherent effectiveness of semi-supervised learning methods. I am curious about what specific contributions the authors have made in this regard.
3. It should be clarified how CVaR is incorporated into the RiskModel, with a more detailed explanation provided. Additionally, an analysis of the impact of relevant hyperparameters should be included.
4. The operation symbols in Equations (5) and (6) are not explained or introduced.
5. The authors should present and analyze the model training curves, such as loss, policy entropy, KL divergence, and other relevant metrics.
6. The font size in Table 1 and Table 2 is too small; it is recommended to adjust it for better readability.

**Questions:**

Please refer to the “Weakness” section for related questions.

---

> ### Author Response · Authors · 2025-11-25
> **Response to Reviewer j9V1; Question1**
>
> Thank you for your review and efforts. Below are our responses to each of your questions. If these address your concerns, we hope you will provide timely feedback.
>
> **Question1**: My main concern is that the proposed method in this paper appears more like a technical implementation strategy, lacking sufficient theoretical support. Although some experimental results are provided, they are not comprehensive enough. I recommend adding experiments under different hyperparameters, such as varying temperatures and learning rates, to better validate the method's effectiveness and robustness.
>
> **Response1**: Thanks for your constructive feedback. The efficacy of our proposed approach has solid theoretical grounding,  because in the setting of preference learning, higher-quality preference labeled data would generally mean higher performance for LLM or VLMs. Our proposed approach can automatically construct high-quality preference data with only a limited label information, effectively reducing the cost of manual annotation. Therefore, we compare it with the advanced models. e.g., Idefics2-8B, Llava1.5-7B, MiniCPM-V 2.6 which similarly serve as reward models. Our evaluation results as shown in Table 1 demonstrate that it performs considerably better than these advanced models.
>
>
> On the theoretical grounding of risk model, the crucial benefit of risk model over a simple binary classifier is that it measures risk based on a distribution, which encodes fluctuation as risk. The necessity of fluctuation risk is inspired by extensive risk research in investment theory. Therefore, the proposed risk model leverages a normal distribution to represent match/mismatch probability. All of our technical designs are centered around how to accurately estimate the normal distribution. Therefore, the methodological complexity is justified.
>
> Specifically, we provide a detailed exposition of the theoretical underpinnings of our modelling approach in Section 3 of the paper. We ingeniously abstract the model's inputs and outputs into a single data pair, which is treated as a random event representing either a match or a mismatch. This random variable follows a Bernoulli distribution, which approximates a normal distribution in scenarios involving multiple samples. As these principles constitute fundamental probability theory, we have refrained from elaborating further. By combining the normal distribution with fundamental venture capital principles, we can calculate the risk value associated with the random variable matching or failing to match.
>
>
> Regarding experiments, it is noteworthy to emphasize that our primary contribution is on constructing a preference dataset closely aligned with ground truth using only a limited amount of labeled data, but not on preference fine-tuning methods. Firstly, we demonstrate by Table 1 that the preference labels generated by SPRA are considerably more accurate than the existing alternatives of large models. Secondly, we demonstrate by Table 3 that using only a limited amount of ground-truth labels, SPRA can achieve highly competitive preference learning compared to the SCL using all the ground-truths.
>
> Additionally, we repeated the experiment five times from start to finish, including risk feature generation, risk model construction, and training. in Appendix E, demonstrating the vigor of our methodology. Due to character limitations, we have listed only a portion of the results here.
> | Data       | Method         | 1      | 2      | 3      | 4      | 5      | Mean±SD        |
> |------------|----------------|--------|--------|--------|--------|--------|----------------|
> | **Animals** | Idefics2-8B     | 43.88  | 46.39  | 46.27  | 45.51  | 45.51  | 45.51 ± 0.90   |
> |            | Llava1.5-7B    | 51.44  | 51.50  | 51.53  | 51.44  | 51.53  | 51.53 ± 0.06   |
> |            | MiniCPM-V 2.6  | 55.20  | 55.03  | 55.74  | 55.35  | 55.05  | 55.27 ± 0.26   |
> |            | **SPRA(Ours)** | **80.01** | **78.04** | **79.23** | **77.29** | **77.15** | **78.73 ± 1.01** |
> | **RealWorld** | Idefics2-8B   | 61.46  | 55.00  | 56.50  | 55.00  | 53.50  | 56.29 ± 2.75   |
> |            | Llava1.5-7B    | 39.52  | 34.29  | 30.00  | 36.67  | 36.19  | 35.33 ± 3.15   |
> |            | MiniCPM-V 2.6  | 59.51  | 61.46  | 65.85  | 62.44  | 63.90  | 62.63 ± 2.15   |
> |            | **SPRA(Ours)** | **73.68** | **74.64** | **66.71** | **69.18** | **69.18** | **70.68 ± 3.00** |
> | **MMbench** | Idefics2-8B     | 59.38  | 60.76  | 57.35  | 58.57  | 60.40  | 59.29 ± 1.24   |
> |            | Llava1.5-7B    | 50.86  | 51.33  | 49.60  | 47.63  | 52.02  | 50.29 ± 1.55   |
> |            | MiniCPM-V 2.6  | 64.82  | 65.59  | 66.82  | 68.05  | 68.82  | 66.82 ± 1.50   |
> |            | **SPRA(Ours)** | **79.68** | **78.93** | **76.21** | **74.90** | **77.43** | **77.43 ± 1.74** |

---

> > ### Author Response · Authors · 2025-12-02
> > **Revision based on Question1**
> >
> > **Q1**.My main concern is that the proposed method in this paper appears more like a technical implementation strategy, lacking sufficient theoretical support. Although some experimental results are provided, they are not comprehensive enough. I recommend adding experiments under different hyperparameters, such as varying temperatures and learning rates, to better validate the method's effectiveness and robustness.
> >
> > **Revision**: 1). We highlighted the theoretical analysis of SPRA in the methodology section at line 179:
> >
> >     “We define the relationship between the input and output of a vision-language model(VLM) as a matching process. As usual, SPRA treats the matching probability of each data pair as a random variable that follows a Bernoulli distribution with parameter $p$, where $p$ follows a $Beta(\alpha, \beta)$ distribution. The $\alpha$ and $\beta$ are the shape parameters of the $Beta$ distribution. From a statistical perspective, the Beta distribution can be approximated by a normal distribution when $\alpha + \beta \geq 10$. Specifically, in the context of the task scenario, $\alpha + \beta$ represents the total number of samples. Our approach utilizes a set of general rules to capture the risk features of a given dataset. Correspondingly, data with distinct risk features exhibit different matching probabilities. This implies that random variables with varying risk features follow different normal distributions, each with its own mean and variance. This is crucial for accurate risk assessment. In the rest of this section, we will present the technical details of SPRA, including risk model construction \& training and risk-based preference finetuning.”
> >
> > 2). We clarified our contributions at line 107:
> >
> >     “The efficacy of our proposed approach has solid theoretical grounding, because in the setting of preference learning, higher-quality preference labeled data would generally mean higher performance for LLM or VLMs. Our proposed approach can automatically construct high-quality preference data with only a limited label information, effectively reducing the cost of manual annotation. In summary, compared with the existing reward models, our proposed SPRA has three key advantages:
> >
> >     - It utilizes only a small number of labeled samples to effectively learn and quantify the relationship between the large model's outputs and human preferences, significantly reducing the burden of manual annotation.
> >     - It measures risk, or preference alignment, by a distribution. Compared with single values predicted by existing reward models, distributions can effectively capture fluctuation risk, whose necessity has been widely accepted by existing risk research.
> >     - It incorporates human-centric priors with a small-sized risk model, making the process of preference finetuning more interpretable and resource-efficient. Compared with existing single reward models, which are usually prone to introducing bias, the proposed risk model can effectively reduce bias by fusing independent outputs from various large models.

---

> ### Author Response · Authors · 2025-11-25
> **Response to Reviewer j9V1; Question2-4**
>
> **Question2**:	The paper compares the proposed SPRA method with SCL in experiments, and the results show that SPRA does not demonstrate a significant performance advantage. Although the authors mention that SPRA uses only a small amount of ground-truth labels compared to SCL, this advantage should primarily be attributed to the inherent effectiveness of semi-supervised learning methods. I am curious about what specific contributions the authors have made in this regard.
>
> **Response2**: Thanks for your insightful comment. First of all, for preference learning, 1-3% improvements can be generally considered substantial for VLM downstream task, e.g., question-answering tasks[1, 2, 3]. Secondly, our primary contribution is not on semi-supervised model optimization, but on constructing a dataset approximating ground truth using SPRA based on a small amount of labeled data. SPRA can significantly reduce manual annotation costs while delivering quality comparable to ground-truth.
>
> It is noteworthy that SCL is a classical method of preference fine-tuning, aiming to validate the quality of constructed preference datasets. While comparing SPRA with SCL, they actually use the same fine-tuning method. The difference is on how to construct preference data, SCL using all the ground-truth labels while SPRA leveraging only a small amount of ground-truth labels.
>
> Analysis of the SCL[1] results indicates that a 3% improvement in model performance is already significant.
>
> | Model                     | RealWorldQA | MMStar | MMBench | SEEDBench | ScienceQA | MMT-Bench | MMMU  | AI2D  | Rank |
> |---------------------------|-------------|--------|---------|-----------|-----------|-----------|--------|--------|------|
> | **LLaVA-V1.5-13B**        |             |        |         |           |           |           |        |        |      |
> | +SFT                     | **56.59**   | 36.33  | 75.16   | **69.32** | 71.88     | 40.08     | 34.88 | 55.95 | 2.75 |
> | +SCL               | 56.03       | **37.60** | **75.40** | 68.56      | **72.16** | **41.16** | **35.87** | **58.92** | **1.38** |
> | **InternLM-XComposer-2-7B** |          |        |         |           |           |           |        |        |      |
> | +SFT                     | 62.03       | 48.90  | **77.80** | **70.20** | 79.50     | **50.20** | 57.95 | 80.66 | 1.75 |
> | +SCL              | **62.23**   | **49.60** | 77.00   | 70.60      | **79.90** | **50.20** | **58.20** | **81.11** | **1.13** |
>
>
> [1] Self-Correction is More than Refinement: A Learning Framework for Visual and Language Reasoning Tasks. ACL Findings (2025)
>
> [2]. SimPO: Simple Preference Optimization with a Reference-Free Reward. NeurIPS (2024)
>
> [3]. β-DPO: Direct Preference Optimization with Dynamic β. NeurIPS (2024)
>
>
> **Question3**: It should be clarified how CVaR is incorporated into the RiskModel, with a more detailed explanation provided. Additionally, an analysis of the impact of relevant hyperparameters should be included.
>
> **Response3**: Thanks for your suggestion. First of all, we need to clarify that we didn’t mention the metric of CVaR in the paper. We instead use the metric of VaR to quantify risk. The use of VaR is aligned with the existing work on risk measure for AI models. Section 3.1.2 provides a detailed explanation of how to compute Var. We will improve presentation to clarify the concept of VaR.
>
>
> $$
> \mathrm{VaR}_{c}(d_i) = F_i^{-1}\left(c;\,\bar{\mu}_i,\,\bar{\sigma}_i^{\,2}\right)
> $$
>
>
> where $d_i$ is a data pair, $F_i^{-1}(\cdot)$ is the quantile function of the normal distribution. $\bar{\mu_i}$ and $\bar{\sigma_i}^2$ denote the mean and variance of the normal distribution that $d_i$ corresponds to. $c$ is set to 90 in our experiments.
>
>
>
> with the algorithm flowchart included in Appendix F.
>
> **Require**:
> - Subset of dataset \(D_s = \{d_1, \dots, d_n\}\)
> - Risk rules \(R = \{r_1, \dots, r_m\}\)
> - Risk features \(F = \{f_1, \dots, f_m\}\)
> - Activation matrix \(A = 0\)
>
> ---
>
> 1. **For** \(i = 1\) to \(n\):
>     1. **For** \(j = 1\) to \(m\):
>         - **If** sample \(d_i\) satisfies feature \(f_j\):
>           → Set \(A_{ij} = 1\)
>         - **Else**
>           → Set \(A_{ij} = 0\)
>     2. **End for**
> 2. **End for**
>
> 3. Compute column-wise mean of activated samples using Eq. (2):
>    → \(\mu = \{\mu_1, \dots, \mu_m\}\)
>
> 4. Generate risk labels \(y\) for \(D_s\).
>
> 5. Construct risk model using Eq. (6).
>
> 6. Optimize risk model using Eq. (8).
>
> 7. **Return** final trained risk model.
>
>
> **Question4**:	The operation symbols in Equations (5) and (6) are not explained or introduced.
>
> **Response4**:Thanks for the comment. Lines 300–311 provide a detailed description of the symbols used in the formula. Certain operators are universally recognized in mathematics, hence no additional textual explanation is provided. We will improve presentation to clarify the mentioned symbols.

---

> > ### Author Response · Authors · 2025-12-02
> > **Revision based on Question2-4**
> >
> > **Q2**: The paper compares the proposed SPRA method with SCL in experiments, and the results show that SPRA does not demonstrate a significant performance advantage. Although the authors mention that SPRA uses only a small amount of ground-truth labels compared to SCL, this advantage should primarily be attributed to the inherent effectiveness of semi-supervised learning methods. I am curious about what specific contributions the authors have made in this regard.
> >
> > **Revision**: 1). At line 107, we revise the description of our contributions to improve clarity. The content is consistent with the description of our contributions in R1(revision based on Q1).
> >
> > 2). We add the following description in Section 4 to make the experimental logic clearer and to emphasize that the primary goal of SPRA is preference prediction rather than model fine-tuning:
> >
> >     “This section evaluates the efficacy of the proposed SPRA by an empirical study on real benchmark datasets. Since the major contribution of SPRA is using a small amount of labeled data to automatically annotate a large volume of unlabeled data, which can then be leveraged to fine-tune preferences of VLMs. Therefore, our experiments have two primary objectives: 1) to demonstrate that the preference labels generated by SPRA are considerably more accurate than the existing alternatives of large reward models; 2) to demonstrate that using only a limited amount of ground-truth labels, SPRA can achieve highly competitive preference learning compared to the classical preference fine-tuning method of SCL using all the ground-truths.”
> >
> >
> > 3). We add an experiment demonstrating the performance improvement obtained by VLMs on a dataset constructed using the same 40% ground-truth data as SPRA. The specific description can be found at line 513 and 522:
> >
> >     “As mentioned above, we have implemented two versions of SCL, SCL-100%, which uses all the ground-truths in the training data of benchmarks, and SCL-40%, which uses the same amount of labeled data as SPRA.”
> >
> > The related description is also at line 522:
> >
> >     “Secondly, it can be observed that if using only 40% of the ground-truth in the training dataset, the performance of SCL is consistently worse than SPRA. This suggests that leveraging a small set of labeled examples to annotate a large amount of unlabeled data is indeed meaningful, as it leads to more substantial performance gains.”
> >
> >
> > **Q3**: The operation symbols in Equations (5) and (6) are not explained or introduced.
> >
> > **Revision**: In line 331, we added the following content to describe the operators:
> > $\cdot$ denotes matrix multiplication, $\circ$ denotes element-wise multiplication
> >
> > **Q4**.The authors should present and analyze the model training curves, such as loss, policy entropy, KL divergence, and other relevant metrics.
> >
> > **Revision**: 1). In line 107 of the introduction, we emphasize that the goal of SPRA is to replace human annotation and thereby reduce labeling costs:
> >
> >
> >     "The efficacy of our proposed approach has solid theoretical grounding,  because in the setting of preference learning, higher-quality preference labeled data would generally mean higher performance for LLM or VLMs. Our proposed approach can automatically construct high-quality preference data with only a limited label information, effectively reducing the cost of manual annotation. "
> >
> > 2). In line 162 of the related work section, we also provide a corresponding discussion.
> >
> >
> >     "Finally, our work in this paper targets preference learning, we therefore use the mainstream DPO solution for preference fine-tuning. It is noteworthy that our primary contribution is on constructing a preference dataset closely aligned with ground truth using only a limited amount of labeled data, but not on preference fine-tuning methods. Therefore, the existing work on preference fine-tuning methods, e.g., DPO, are orthogonal to ours."

---

> ### Author Response · Authors · 2025-11-25
> **Response to Reviewer j9V1; Question5-6**
>
> **Question5**: The authors should present and analyze the model training curves, such as loss, policy entropy, KL divergence, and other relevant metrics.
>
> **Response5**: Thanks for the comment. It is worthy to emphasize that our primary contribution is not on semi-supervised model optimization, but on constructing a dataset approximating ground truth using SPRA based on a small amount of labeled data. SPRA can significantly reduce manual annotation costs while delivering quality equivalent to ground truth.
>
> As a matter of fact, the proposed SPRA uses the same preference fine-tuning method as SCL. As observed in our experiments, both SCL and SPRA is highly stable. Running multiple rounds with different configurations still yields stable outputs.
>
> **Question6**:The font size in Table 1 and Table 2 is too small; it is recommended to adjust it for better readability.
>
> **Response6**: Thank you for the comment; we shall adjust the font size as suggested.

---

> > ### Author Response · Authors · 2025-12-02
> > **Revision based on 6**
> >
> > **Q6**.	The font size in Table 1 and Table 2 is too small; it is recommended to adjust it for better readability.
> >
> > **Revision**: We enlarge the font size in Tables 1 and 2.

---

### Official Review · Reviewer_ezGu · 2025-11-01

**Soundness:** 3
**Presentation:** 2
**Contribution:** 3
**Rating:** 6
**Confidence:** 2

**Summary:**

The paper introduces a new framework for aligning vision-language models (VLMs) with human preferences using limited labeled data. Instead of relying on costly reward models trained from large-scale human feedback, SPRA models preference alignment as a risk estimation problem: outputs that deviate from human priors are treated as high-risk. The method constructs risk features—both statistical (confidence-based) and generative (consistency-based)—to quantify how well a model’s response conforms to interpretable human-like rules. A small risk model then aggregates these features into a continuous “Value-at-Risk” score that ranks outputs by preference likelihood. This enables efficient fine-tuning via Direct Preference Optimization (DPO). Experiments across classification and VQA benchmarks show that SPRA achieves accuracy comparable to or better than fully supervised methods while requiring far fewer human annotations.

**Strengths:**

originality: The paper introduces SPRA – a framework to align VLM with human preference leveraging a light-weighted risk model. The risk model is trained on a set of risk features marking deviation from human priors. This is a novel idea of designing a set of simple oracle rewards for VLM alignment rather than user large model judges, making preference alignment more cost-efficient.
quality & clarity: The paper provides quite clear derivation of different risk features, detailed experimental results and ablation results in the appendix. Overall writing and figures are clear except for the point mentioned in weakness.
significance: The developed method labels preference dataset for VLM alignment with much reduced human or compute resource as it uses a light-weight risk model. It also shows better grasp of the underlying ground truth label as well as competitive performance in align generation policy at downstream.

**Weaknesses:**

1. Figure clarity: Figure 2 the arrow "Bert-score" occludes part of the illustrative text; Figure 3 contains wrong spelling "Optiions" and the workflow is confusing.
2. Section 4.2 Table 2: In Table 1, the authors show that their method SPAR largely improve the prediction accuracy of the ground truth label. However, in using SPAR for fine-tuning, the gain on the downstream tasks are much less. Could authors provide more intuition regarding this?
3. Section 3.1.1 risk feature generation: I understand that the authors try to find a set of simplified oracles to define what a human would call "risk." However, I need more motivation and explanation on the statistical risk feature part. Take entropy as an example, if the model generate high entropy tokens it doesn't necessarily indicate the model is not aligned. It could be your input data is low quality? Given an ambiguous image e.g. the model naturally would have high uncertainty in generation, not necessarily because it's policy is not aligned.

**Questions:**

1. Regarding weakness 1: it'd be helpful to improve on figure clarity.
2. See weakness 2.
3. See weakness 3.

---

> ### Author Response · Authors · 2025-11-25
> **Response to Reviewer ezGu; Question1-2**
>
> **Question 1**:	Figure clarity: Figure 2 the arrow "Bert-score" occludes part of the illustrative text; Figure 3 contains wrong spelling "Optiions" and the workflow is confusing.
>
> **Response 1**: Thank you for the comment. We will improve the presentation based on your feedback.
>
> **Question 2**:   Section 4.2 Table 2: In Table 1, the authors show that their method SPAR largely improve the prediction accuracy of the ground truth label. However, in using SPAR for fine-tuning, the gain on the downstream tasks are much less. Could authors provide more intuition regarding this?
>
>
> **Response 2**: Thanks for the insightful comment. For preference learning, 1-3% improvements can be generally considered substantial for VLM downstream task, e.g., question-answering tasks[1, 2, 3]. For instance, for the Q&A task, we list below the improvements achieved by the classical SCL based on DPO on popular datasets, demonstrating that the enhancements of the model in our paper also fall within the effective range. Furthermore, it is noteworthy to emphasize that our primary contribution is on constructing a preference dataset closely aligned with ground truth using only a limited amount of labeled data, but not on preference fine-tuning. The evaluation results show that SPRA yields outcomes comparable to the SCL using all the ground-truths, thereby significantly reducing the labor cost associated with manual preference annotation.
>
>
> | Model                     | RealWorldQA | MMStar | MMBench | SEEDBench | ScienceQA | MMT-Bench | MMMU  | AI2D  | Rank |
> |---------------------------|-------------|--------|---------|-----------|-----------|-----------|--------|--------|------|
> | **LLaVA-V1.5-13B**        |             |        |         |           |           |           |        |        |      |
> | +SFT                     | **56.59**   | 36.33  | 75.16   | **69.32** | 71.88     | 40.08     | 34.88 | 55.95 | 2.75 |
> | +SCL              | 56.03       | **37.60** | **75.40** | 68.56      | **72.16** | **41.16** | **35.87** | **58.92** | **1.38** |
> | **InternLM-XComposer-2-7B** |          |        |         |           |           |           |        |        |      |
> | +SFT                     | 62.03       | 48.90  | **77.80** | **70.20** | 79.50     | **50.20** | 57.95 | 80.66 | 1.75 |
> | +SCL              | **62.23**   | **49.60** | 77.00   | 70.60      | **79.90** | **50.20** | **58.20** | **81.11** | **1.13** |
>
> [1]. SimPO: Simple Preference Optimization with a Reference-Free Reward. NeurIPS (2024)
>
> [2]. β-DPO: Direct Preference Optimization with Dynamic β. NeurIPS (2024)
>
> [3]. Self-Correction is More than Refinement: A Learning Framework for Visual and Language Reasoning Tasks. ACL Findings (2025)

---

> > ### Author Response · Authors · 2025-12-02
> > **Revision based on Question1-2**
> >
> > **Q1**.Figure clarity: Figure 2 the arrow "Bert-score" occludes part of the illustrative text; Figure 3 contains wrong spelling "Optiions" and the workflow is confusing.
> >
> > **Revision**: We revised Figures 2 and 3 to make the workflow clearer. In addition to correcting the typos shown in the figures, we also reviewed the entire manuscript for typographical errors. All identified typos have been corrected and highlighted in red.
> >
> > **Q2**.Section 4.2 Table 2: In Table 1, the authors show that their method SPAR largely improve the prediction accuracy of the ground truth label. However, in using SPAR for fine-tuning, the gain on the downstream tasks are much less. Could authors provide more intuition regarding this?
> >
> > **Revision**:  1). We added new references at line 498 showing that fine-tuned models—whether trained on SPRA-labeled or on ground-truth–labeled preference datasets, both results are shown as reliable on verifiable tasks:
> >
> >       “It is worth noting that our experiments only applied DPO to the model without performing supervised fine-tuning(SFT), in order to align with the experimental results in SCL. With only preference learning, pretrained large models are inherently difficult to improve significantly. For instance, in verifiable tasks (e.g., Q&A), leading optimization methods like SimPO(Meng et al., 2024) and β-DPO(Wu et al., 2024) ave demonstrated that an improvement of nearly 3% is typically considered effective.”
> > 2). we highlighted our main theme by adding explanation in Section 2 “RELATED WORK”, It is marked in red:
> >
> >      “Finally, our work in this paper targets preference learning, we therefore use the mainstream DPO solution for preference fine-tuning. It is noteworthy that our primary contribution is on constructing a preference dataset closely aligned with ground truth using only a limited amount of labeled data, but not on preference fine-tuning methods. Therefore, the existing work on preference fine-tuning methods, e.g., DPO, are orthogonal to ours.”
> >
> > 3). We also add the following description in Section 4 to make the experimental logic clearer and to emphasize that the primary goal of SPRA is preference prediction rather than model fine-tuning:
> >
> >      “This section evaluates the efficacy of the proposed SPRA by an empirical study on real benchmark datasets. Since the major contribution of SPRA is using a small amount of labeled data to automatically annotate a large volume of unlabeled data, which can then be leveraged to fine-tune preferences of VLMs. Therefore, our experiments have two primary objectives: 1) to demonstrate that the preference labels generated by SPRA are considerably more accurate than the existing alternatives of large reward models; 2) to demonstrate that using only a limited amount of ground-truth labels, SPRA can achieve highly competitive preference learning compared to the classical preference fine-tuning method of SCL using all the ground-truths.”

---

> ### Author Response · Authors · 2025-11-25
> **Response to Reviewer ezGu; Question3**
>
> **Question 3**: Section 3.1.1 risk feature generation: I understand that the authors try to find a set of simplified oracles to define what a human would call "risk." However, I need more motivation and explanation on the statistical risk feature part. Take entropy as an example, if the model generate high entropy tokens it doesn't necessarily indicate the model is not aligned. It could be your input data is low quality? Given an ambiguous image e.g. the model naturally would have high uncertainty in generation, not necessarily because it's policy is not aligned.
>
> **Response 3**: Thanks for your review. A risk feature is supposed to indicate the uncertainty of an output from one perspective. However, using individual risk features to describe the risks inherent in a large model's output is inadequate. We have introduced various risk features capable of indicating the presence of risk in the model's output. Regarding the entropy mentioned by the reviewer, our understanding is as follows: The ultimate output of a large model constitutes a probability distribution, with tokens sampled from this distribution (typically selecting the index with the highest probability). Viewing the model from the statistical perspective: if the top-n probabilities within its output distribution are nearly identical, this indicates the model lacks confidence in its current answer. Conversely, if one probability significantly exceeds the others, it signifies the model is highly confident in its current answer and deems it correct. Therefore, statistically speaking, the metric of entropy can effectively indicate the model's belief in its answer. Hence, we incorporate entropy as a risk indicator to assess whether the model's output carries potential hazards. The selection of entropy metrics is detailed in Appendix C.1.
>
>
> Regarding the connection you mentioned between ambiguous images and large model outputs, this is precisely why we introduce multiple risk rules instead of just one. In the risk rules we define, if an image is ambiguity, most risk rules become ineffective. For example, object detection within the rules and question-answering based on detection results. Most of our rules are based on the primary objects within an image. An ambiguous image indicates that the primary object recognition has failed, resulting in most risk features being set to 1. This consequently increases the likelihood that the risk model will output a high-risk result. Human preferences are human-centric. Even if a model's output does not align with objective reality, if it provides the answer users want to hear, then that answer is considered consistent with human preferences. However, this does not necessarily imply its correctness. For classification tasks, if an image shows a Dalmatian dog, it is classified as “dog.” Only after aligning with preferences will the model provide the answer users expect—Dalmatian dog, not a generic dog.

---

> > ### Comment · Reviewer_ezGu · 2025-11-25
> >
> > Thanks for authors' response to my questions. I see that the authors promise to change result presentations but haven't actually updated the paper pdf. Additionally, the authors have provided new results in responding to other reviewers' comment. I find it hard to imagine the paper's content without actually seeing the changes. I would like to see the updated version of the paper.

---

> > > ### Author Response · Authors · 2025-11-26
> > >
> > > Thank you for your response and for your positive recognition of our work. We plan to collect feedback from all reviewers before updating both the new and existing content in the paper. Your suggestions, along with the comments from other reviewers, have already been incorporated into the revised version of the manuscript. As for the table font sizes, we will adjust them in the final stage to ensure that the paper stays within the page limit. More details are provided in Appendices G and H.
> > >
> > >
> > > Thank you again.

---

> > ### Author Response · Authors · 2025-12-02
> > **Revision based on Question3**
> >
> > **Q3**.Section 3.1.1 risk feature generation: I understand that the authors try to find a set of simplified oracles to define what a human would call "risk." However, I need more motivation and explanation on the statistical risk feature part. Take entropy as an example, if the model generate high entropy tokens it doesn't necessarily indicate the model is not aligned. It could be your input data is low quality? Given an ambiguous image e.g. the model naturally would have high uncertainty in generation, not necessarily because it's policy is not aligned.
> >
> > **Revision**: 1). Due to space limitations, we provide the explanation of the statistical risk features in Appendix C.1. The detailed content is as follows:
> >
> >      “We designed multiple rules to evaluate the risks associated with large-model outputs. In the risk rules we define, if f the input is abnormal or the image is ambiguity, most risk rules become ineffective. For example, object detection within the rules and question-answering based on detection results. Most of our rules are based on the primary objects within an image. A ambiguous image indicates that the primary object recognition has failed, resulting in most risk features being set to 1. This consequently increases the likelihood that the risk model will output a high-risk result.”
> >
> >
> > 2). At line 279, we added a discussion of these risk features to clarify the stability of the SPRA approach:
> >
> >     “A risk feature is supposed to indicate the uncertainty of an output from one perspective. However, using any individual risk feature to describe the risk inherent in a large model's output is usually inadequate. On the other hand, any single risk feature (or reward model) would be prone to introducing bias. Therefore, our proposed approach leverages various risk features, which are supposed to be constructed based on various outputs from different large models. It can be expected that fusing outputs from different large models would effectively reduce preference bias.”

---

### Meta-Review · Area_Chair_67a2 · 2026-01-10

**Summary:**

This paper proposes SPRA, a semi-supervised preference-learning framework for multimodal models that tries to reduce reliance on large-scale human preference annotations by constructing a lightweight “risk model” from a small labeled set. The method hand-designs and extracts risk features (intended to encode human priors), assumes a probabilistic form over risk, and uses a VaR-style objective to score and rank candidate outputs so that lower-risk responses are treated as preferred for downstream preference optimization (e.g., DPO). Experiments report improved preference-prediction accuracy and downstream fine-tuning gains on several VLM benchmarks, but the approach appears to depend heavily on the chosen priors/features and distributional assumptions, and the “risk label” is tied to matching ground truth rather than directly capturing human preference.

**Reviewer Concerns:**

Initially, the reviewers raised several concerns, including the significance of results, confusing presentations, insufficient experiments (more recent models and baselines), and computational cost. Despite the authors’ rebuttal, some concerns still remained as explicitly denoted by the reviewer response (R8uo).

**Reviewer Scores:**

Initially, the reviewers' scores were (6,4,4,2), leaning to the rejection (avg: 4.0). Despite the authors providing the extensive rebuttal, some reviewers did not respond to this. After carefully reading all the reviews and rebuttals by the authors, AC believes that some concerns, especially of bPAR and R8uo, are not resolved sufficiently and hence reviewers would still lean to the rejection on average (6 or 4,4 or 6,4,2). The detailed reasons are as follow:

ezGu: 6 -> 6 or 4
- If the reviewer is satisfied with the actual change, the score would remain. If not, the score would decrease as reflected in his/her response.

j9V1: 4 -> 4 or 6
- Most concerns were resolved, but some concerns still remain; for example, the model training curve is not provided despite the reviewer’s request. If the reviewer accepts the authors’ claim (their contribution is dataset construction, not training itself), the score could be increased. If not, the score would remain; regarding this, AC recommends the authors to change the title to clarify the contribution since the current title is more like a training method.

bPAR: 4 -> 4
- The responses for some requests like inclusion of recent models and SFT baseline were not provided. While the authors mentioned that SFT is not comparable in the context of preference learning, it is not true as SFT with preferred positive data is considerable baseline.

R8uo: 2 -> 2
- As denoted in the recursive responses by R8uo, the authors’ rebuttal failed to successfully address his/her concerns such as the necessity of methodological complex in perspective of empirical results and computational costs.

---

### Decision · Program_Chairs · 2026-01-26

Reject